# Systematic Visual Reasoning through Object-Centric Relational Abstraction

**Taylor W. Webb\*†**
Department of Psychology
University of California, Los Angeles
Los Angeles, CA
taylor.w.webb@gmail.com

**Shanka Subhra Mondal\*†**
Department of Electrical and Computer Engineering
Princeton University
Princeton, NJ
smondal@princeton.edu

**\* Equal Contribution**

**Jonathan D. Cohen**
Princeton Neuroscience Institute
Princeton University
Princeton, NJ
jdc@princeton.edu

## Abstract

Human visual reasoning is characterized by an ability to identify abstract patterns from only a small number of examples, and to systematically generalize those patterns to novel inputs. This capacity depends in large part on our ability to represent complex visual inputs in terms of both objects and relations. Recent work in computer vision has introduced models with the capacity to extract object-centric representations, leading to the ability to process multi-object visual inputs, but falling short of the systematic generalization displayed by human reasoning. Other recent models have employed inductive biases for relational abstraction to achieve systematic generalization of learned abstract rules, but have generally assumed the presence of object-focused inputs. Here, we combine these two approaches, introducing Object-Centric Relational Abstraction (OCRA), a model that extracts explicit representations of both objects and abstract relations, and achieves strong systematic generalization in tasks (including a novel dataset, CLEVR-ART, with greater visual complexity) involving complex visual displays.

## 1  Introduction

When presented with a visual scene, human reasoners have a capacity to identify not only the objects in that scene, but also the relations between objects, and the higher-order patterns formed by multiple relations. These abilities are fundamental to human visual intelligence, enabling the rapid induction of abstract rules from a small number of examples, and the systematic generalization of those rules to novel inputs [32, 25, 13, 27]. For example, when presented with the image in Figure 1, one can easily determine that the objects in the top and bottom rows are both governed by a common abstract pattern (ABA). Furthermore, one could easily then generalize this pattern to *any* potential shapes, even those that one has never observed before. Indeed, this is arguably what it means for a pattern to be considered *abstract*.

37th Conference on Neural Information Processing Systems (NeurIPS 2023).

† Co-first author order is arbitrary and may be swapped when citing this work.

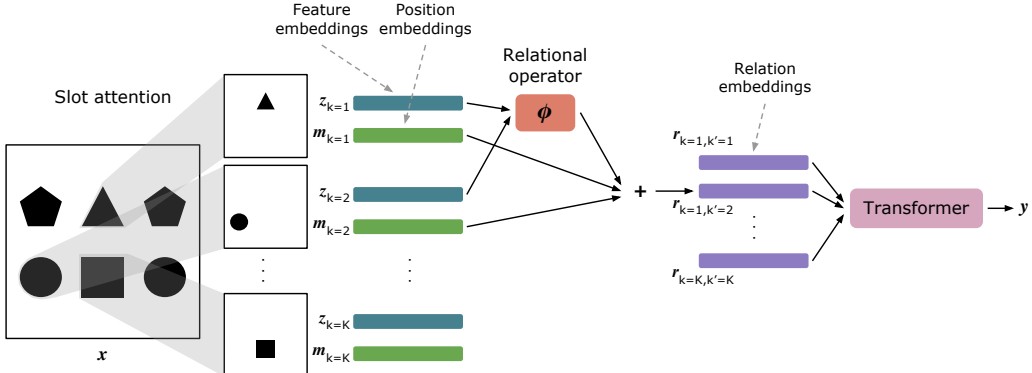

Figure 1: **Object-Centric Relational Abstraction (OCRA).** OCRA consists of three core components, responsible for modeling objects, relations, and higher-order relations. First, given a visual input $x$, slot attention extracts object-centric representations, consisting of factorized feature embeddings $z_{k=1...K}$ and position embeddings $m_{k=1...K}$. Second, pairwise relation embeddings $r_{kk'}$ are computed by first passing each pair of feature embeddings to a relational operator $\phi$, and then additively incorporating their position embeddings. This allows visual space to serve as an indexing mechanism for visual relations. Finally, relational embeddings are passed to a transformer to model the higher-order patterns formed by multiple relations.

Though this task appears effortless from the perspective of human reasoning, it exemplifies the type of problems that pose a significant challenge for standard neural network methods. Rather than inducing an abstract representation of the underlying rules in a given task, neural networks tend to overfit to the specific details of the examples observed during training, thus failing to generalize those rules to novel inputs [26, 3, 33].

To address this, a number of neural network methods have recently been proposed that incorporate strong inductive biases for *relational abstraction* [49, 51, 22, 1], also referred to as a 'relational bottleneck' [50]. The key insight underlying these approaches is that abstract visual patterns are often characterized by the *relations* between objects, rather than the features of the objects themselves. For example, the pattern seen in Figure 1 is characterized by the relation between the shape and position of the objects. By constraining architectures to process visual inputs in terms of relations between objects, these recent approaches have achieved strong systematic (i.e., out-of-distribution) generalization of learned abstract rules, given only a small number of training examples [51, 22, 1].

However, a major assumption of these approaches is that relations are computed between *objects*. Accordingly, these methods have generally relied on inputs consisting of pre-segmented visual objects, rather than multi-object scenes. Furthermore, these methods can sometimes be brittle to task-irrelevant visual variation, such as spatial jitter or Gaussian noise [43]. It is thus unclear whether and how these approaches might scale to complex and varied real-world settings.

In this work, we take a step toward the development of relational reasoning algorithms that can operate over more realistic visual inputs. To do so, we turn to object-centric representation learning [15, 6, 28, 12], a recently developed approach in which visual scenes are decomposed into a discrete set of object-focused latent representations, without the need for veridical segmentation data. Though this approach has shown success on a number of challenging visual tasks [10, 39, 30], it has not yet been integrated with the strong relational inductive biases that have previously been shown to enable systematic, human-like generalization [49, 51, 22, 1]. Here, we propose Object-Centric Relational Abstraction (OCRA), a model that combines the strengths of these two approaches. Across a suite of visual reasoning tasks (including a novel dataset, CLEVR-ART, with greater visual complexity), we find that OCRA is capable of systematically generalizing learned abstract rules from complex, multi-object visual displays, significantly outperforming a number of competitive baselines in most settings.

## 2 Approach

Figure 1 shows a schematic depiction of our approach. OCRA consists of three core components, each responsible for a different element of the reasoning process. These three elements can be summarized as 1) extraction of object-centric representations, 2) computation of pairwise relational embeddings, and 3) processing of higher-order relational structures – patterns across multiple relations – in order to identify abstract rules. We describe each of these three components in the following sections.

### 2.1 Object-centric representation learning

OCRA employs slot attention [28] to extract object-centric representations from complex, multi-object visual inputs. Given an image $x$, the goal of slot attention is to extract a set of latent embeddings (i.e., *slots*), each of which captures a focused representation of a single visual object, typically without access to ground truth segmentation data.

To do so, the image is first passed through a series of convolutional layers, generating a feature map $feat \in \mathbb{R}^{H \times W \times D}$. Positional embeddings $pos \in \mathbb{R}^{H \times W \times D}$ are then generated by passing a 4d position code (encoding the four cardinal directions) through a separate linear projection. Feature and position embeddings are additively combined, and then passed through a series of 1x1 convolutions. This is then flattened, yielding $inputs \in \mathbb{R}^{N \times D}$ (where $N = H \times W$), over which slot attention is performed.

The set of $K$ $slots \in \mathbb{R}^{K \times D}$ is then randomly initialized (from a shared distribution with learned parameters), and a form of transformer-style cross-attention is employed to attend over the pixels of the feature map. Specifically, each slot emits a query $q(slots) \in \mathbb{R}^{K \times D}$ (through a linear projection), and each location in the feature map emits a key $k(inputs) \in \mathbb{R}^{N \times D}$ and a value $v(inputs) \in \mathbb{R}^{N \times D}$. For each slot, a query-key attention operation is then used to generate an attention distribution over the feature map $attn = \text{softmax}(\frac{1}{\sqrt{D}} k(inputs) \cdot q(slots)^{\top})$ and a weighted mean of the values $updates = attn \cdot v(inputs)$ is used to update the slot representations using a Gated Recurrent Unit [8], followed by a residual MLP. This process is repeated for $T$ iterations, leading to a competition between the slots to account for each pixel in the feature map.

For the purposes of computing relational embeddings, we employed a factorized object representation format, in which position is represented distinctly from other object features. For a given input image, after performing $T$ iterations of slot attention, we used the final attention map $attn_T \in \mathbb{R}^{K \times N}$ to compute feature- and position-specific embeddings for each slot:

$$z_k = attn_T \, \text{flatten}(feat)[k] \tag{1}$$

$$m_k = attn_T \, \text{flatten}(pos)[k] \tag{2}$$

where $feat$ and $pos$ represent the feature- and position-embedding maps (prior to being combined). As will be seen in the following section, this factorized representation allows visual space to serve as an indexing mechanism for pairwise relations, mirroring the central role of spatial organization in human visual reasoning [29].

Although slot attention can in principle be learned end-to-end in the service of a specific downstream reasoning task, here we chose to pre-train slot attention using an unsupervised objective. This decision reflects the assumption that human-like visual reasoning does not occur in a vacuum, and visual representations are generally not learned from scratch when performing each new task. Instead, visual object representations are shaped by a wealth of non-task-specific prior experience. To model the effects of this prior experience, we pre-trained slot attention to reconstruct randomly generated multi-object displays, using a slot-based autoencoder framework (more details in Section 4.3.1). Importantly, for experiments testing generalization of abstract rules to novel objects, we ensured that these objects did not appear in the pre-training data for slot attention.

### 2.2 Relational embeddings

After extracting object embeddings, we employed a novel relational embedding method, designed to allow OCRA to abstract over the perceptual features of individual objects (and thus systematically

generalize learned rules to inputs with novel perceptual features). This embedding method consisted of two steps. First, we applied a relational operator $\phi$ to compute the pairwise relation between two feature embeddings $\boldsymbol{z}_k$ and $\boldsymbol{z}_{k'}$. In the present work, we define this operator as:

$$\phi(\boldsymbol{z}_k, \boldsymbol{z}_{k'}) = (\boldsymbol{z}_k \boldsymbol{W_z} \cdot \boldsymbol{z}_{k'} \boldsymbol{W_z}) \boldsymbol{W_r} \tag{3}$$

where $\boldsymbol{z}_k$ and $\boldsymbol{z}_{k'}$ are first projected through a shared set of linear weights $\boldsymbol{W_z} \in \mathbb{R}^{D \times D}$, and the dot product between these weighted embeddings is then projected back out into the original dimensionality using another set of weights $\boldsymbol{W_r} \in \mathbb{R}^{1 \times D}$. Using dot products to model pairwise relations introduces a layer of abstraction between the perceptual features of the objects and the downstream reasoning process, in that the dot product *only* represents the relations between objects, rather than those objects' individual features. We hypothesized that this relational bottleneck would lead to improved generalization to inputs with novel perceptual features. Additionally, the shared projection weights ($\boldsymbol{W_z}$) enable the model to learn which features are relevant for a given task, and to only compute relations between those features.

After applying the relational operator, we then compute the relational embedding for objects $k$ and $k'$ as:

$$\boldsymbol{r}_{kk'} = \phi(\boldsymbol{z}_k, \boldsymbol{z}_{k'}) + \boldsymbol{m}_k \boldsymbol{W_m} + \boldsymbol{m}_{k'} \boldsymbol{W_m} \tag{4}$$

where the position embeddings $\boldsymbol{m}_k$ and $\boldsymbol{m}_{k'}$ are each projected through a shared set of linear weights[1] $\boldsymbol{W_m} \in \mathbb{R}^{D \times D}$ and added to the output of $\phi$. This allows the relation between objects $k$ and $k'$ to be indexed by their spatial position, thus endowing OCRA with an explicit variable-binding mechanism.

### 2.3   Processing of higher-order relations

After computing all $K^2$ pairwise relational embeddings, we then pass these embeddings to a reasoning process to extract higher-order relations – patterns formed between multiple relations. This is an essential component of visual reasoning, as abstract rules, such as the one displayed in Figure 1, are typically defined in terms of higher-order relations. We model these higher-order interactions using a transformer [45]:

$$\boldsymbol{y} = \text{Transformer}(\boldsymbol{r}_{k=1,k'=1}...\boldsymbol{r}_{k=K,k'=K}) \tag{5}$$

where the inputs are formed by the set of all pairwise relational embeddings $\boldsymbol{r}_{kk'}$ (for all $k = 1...K$ and all $k' = 1...K$), and $\boldsymbol{y}$ is a task-specific output. Because we employ a symmetric relational operator in the present work, we do not in practice present the full matrix of pairwise comparisons, but instead limit these comparisons to those in the upper triangular matrix.

## 3   Related Work

A number of methods for annotation-free object segmentation have recently been proposed [15, 6, 28, 12, 11, 20], including the slot attention method employed by our proposed model [28]. Though the details of these methods differ, they share the general goal of decomposing visual scenes into a set of latent variables corresponding to objects, without access to ground truth segmentation data. Building on the success of these object-encoding methods, some recent work has developed object-centric approaches to visual reasoning that combine slot-based object representations with transformer-based architectures [10, 39, 52, 30]. This work has yielded impressive performance on challenging computer vision tasks, such as question answering from video [10], or complex multi-object visual analogy problems [30], but has thus far been limited to generalization to similar problem types following extensive task-specific training.

In the present work, our goal was to develop a model capable of human-like systematic generalization in abstract reasoning tasks. Our proposed approach builds on the insights of other recent work in this

---

[1]Note that the linear nature of this transformation is important: a nonlinear transformation of the position embeddings (e.g., via a MLP) could result in the extraction of implicit shape information, and therefore break the relational bottleneck.

area, most notably including the *Emergent Symbol Binding Network (ESBN)* [51] and *CoRelNet* [22] architectures, both of which employed the concept of a relational bottleneck in different ways. In the ESBN architecture, separate control and perception pathways interact indirectly through a key-value memory, allowing the control pathway to learn representations that are abstracted over the details represented in the perceptual pathway. Kerg et al. [22] proposed that the ESBN's capacity for abstraction was due to the implicit encoding of relational information in its memory operations, which are mediated by a dot-product-based retrieval operation. Kerg et al. also proposed a novel architecture, CoRelNet, that makes this inductive bias more explicit. The CoRelNet architecture operates by explicitly computing the full matrix of relations (based on dot products) between each pair of objects in its input, and then passing these relations to an MLP for further processing. Both ESBN and CoRelNet displayed strong systematic generalization of learned abstract rules from a small set of examples. Importantly, however, both of these architectures depended on the presence of pre-segmented objects as input. A primary contribution of the present work is to combine these inductive biases for relational abstraction (i.e., the relational bottleneck) with object-centric encoding mechanisms, and to introduce the use of a spatial indexing mechanism that allows the model to apply these inductive biases to spatially embedded, multi-object visual inputs.

In addition to lacking object encoding mechanisms, it was recently shown that the ESBN can sometimes be overly brittle to task-irrelevant visual variation. Vaishnav and Serre [43] evaluated the ESBN and other reasoning architectures on abstract visual reasoning problems similar to those evaluated by Webb et al. [51], but with the introduction of either spatial jitter or Gaussian noise, finding that this significantly impaired performance in the ESBN (as well as other reasoning architectures, such as the transformer [45] and relation net [35]). Vaishnav and Serre also introduced a novel architecture, GAMR, that was more robust to these sources of noise, and able to solve problems based on multi-object inputs, likely due to the use of a novel guided attention mechanism. But despite outperforming ESBN and other architectures in this more challenging visual setting, GAMR nevertheless fell short of systematic generalization in these tasks, displaying a significant drop in performance in the most difficult generalization regimes. Here, we directly confront this challenge, demonstrating that OCRA is capable of systematic generalization comparable to that displayed by ESBN and CoRelNet, but in more complex visual settings.

It is also worth mentioning that other neural network architectures have been proposed that incorporate relational inductive biases in various ways [35, 4, 38, 34, 37, 5], some of which also incorporate object-centric representations [48, 44, 19, 53, 2, 24, 40, 46, 14, 41]. Relative to this previous work, our approach is distinguished by the inclusion of a stronger inductive bias toward relational *abstraction* – that is, a relational bottleneck – for promoting the development of relational representations that fully abstract over the details of individual objects [51, 22, 1]. To empirically evaluate the importance of the relational bottleneck, we compare our approach with a set of baseline models that capture the key computational properties of previous models, including a slot-transformer baseline that combines slot attention with a transformer reasoning module [10, 39, 52, 30], and a slot-interaction-network baseline that combines slot attention with an interaction-network reasoning module [48, 44, 19, 53, 2, 24, 40, 46, 14, 41].

To summarize our contributions relative to previous work, our proposed model includes:

1. A novel object representation format that is factorized into distinct feature and position embeddings (Equations 1 and 2), enabling a form of explicit variable-binding.

2. A novel relational embedding method that implements a relational bottleneck (Equations 3 and 4).

3. An architecture that combines these elements with preexisting components (slot attention and transformers) in a novel manner to support systematic visual reasoning from complex visual displays, including a novel dataset CLEVR-ART with greater visual complexity.

## 4   Experiments

### 4.1   Datasets

We evaluated OCRA on two challenging visual reasoning tasks, ART [51] and SVRT [13], specifically designed to probe systematic generalization of learned abstract rules, as well as a novel dataset, CLEVR-ART, involving more complex visual inputs (Figure 2).

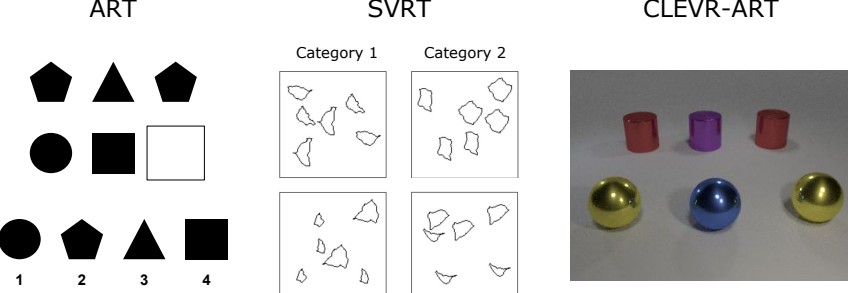

Figure 2: **Abstract visual reasoning tasks.** Three datasets used to evaluate systematic visual reasoning. Abstract Reasoning Tasks (ART) consist of four separate visual reasoning tasks: same/different, relational-match-to-sample, distribution-of-3, and identity rules (pictured). Systematic generalization is studied by training on problems involving a small number of objects (as few as 5), and testing on heldout objects. Synthetic Visual Reasoning Test (SVRT) consists of 23 tasks, each defined by separate abstract rule. Rules are defined based on either spatial relations (SR) or same/different relations (SD; pictured). In example problem, category 1 always contains three sets of two identical objects, category 2 always contains two sets of three identical objects. CLEVR-ART is a novel dataset consisting of ART problems rendered using more realistic 3D shapes, based on the CLEVR dataset.

**ART.** The Abstract Reasoning Tasks (ART) dataset was proposed by Webb et al. [51], consisting of four visual reasoning tasks, each defined by a different abstract rule (Figure S1). In the 'same/different' task, two objects are presented, and the task is to say whether they are the same or different. In the 'relational match-to-sample' task, a source pair of objects is presented that either instantiates a 'same' or 'different' relation, and the task is to select the pair of target objects (out of two pairs) that instantiates the same relation. In the 'distribution-of-3' task, a set of three objects is presented in the first row, and an incomplete set is presented in the second row. The task is to select the missing object from a set of four choices. In the 'identity rules' task, an abstract pattern is instantiated in the first row (ABA, ABB, or AAA), and the task is to select the choice that would result in the same relation being instantiated in the second row.

The primary purpose of this dataset was to evaluate strong systematic generalization of these rules. Thus, each task consists of a number of generalization regimes of varying difficulty, defined by the number of unique objects used to instantiate the rules during training (out of a set of 100 possible objects). In the most difficult regime ($m = 95$), the training set consists of problems that are created using only 5 objects, and the test problems are created using the remaining 95 objects, whereas in the easiest regime ($m = 0$), both training and test problems are created from the complete set of 100 objects (though the arrangement of these objects is distinct, such that the training and test sets are still disjoint). The most difficult generalization regime poses an especially difficult test of systematic generalization, as it requires learning of an abstract rule from a very small set of examples (details shown in Table S1), with little perceptual overlap between training and test.

As originally proposed [51], the ART dataset involved pre-segmented visual objects. Here, we investigated a version of this task involving multi-object visual displays (see Supplementary Section S2 for details). We also applied random spatial jitter (random translation of up to 5 pixels in any direction) to each object, as this was previously found to significantly impair performance in some previous reasoning models [42].

**SVRT.** The Synthetic Visual Reasoning Test (SVRT), proposed by Fleuret et al. [13], consists of 23 binary classification tasks, each defined by a particular configuration of relations. The tasks can be broadly divided into two categories: those that are based primarily on same/different relations ('SD'; tasks 7, 21, 5, 19, 6, 17, 20, 1, 13, 22, 16; example shown in Figure S3(a)), and those that are primarily based on spatial relations ('SR'; tasks 12, 15, 14, 9, 23, 8, 10, 18, 4, 3, 11, 2; example shown in Figure S3(b)). As with previous work [43], we limit training to a relatively small number of examples (either 500 or 1000 per task).

**CLEVR-ART.** We created a novel dataset based on ART using realistically rendered 3D shapes from CLEVR [21] (Figure S4). We focused on two specific visual reasoning tasks: relational-match-

to-sample and identity rules. Problems were formed from objects of three shapes (cube, sphere, and cylinder), three sizes (small, medium, and large), eight colors (gray, red, blue, green, brown, purple, cyan, and yellow) and two textures (rubber and metal). To test systematic generalization, the training set consisted of problems using only small and medium sized rubber cubes in four colors (cyan, brown, green, and gray) whereas the test set consisted only of problems using large sized metal spheres and cylinders in four distinct colors (yellow, purple, blue, and red). The training and test sets thus involved completely disjoint perceptual features, similar to the original ART.

## 4.2 Baselines

We compared our model to results from a set of baseline models evaluated by Vaishnav and Serre [43], including their GAMR architecture, ResNet50 [17], and a version of ResNet that employed self-attention (Attn-ResNet) [42]. To the best of our knowledge, these baselines represent the best performing models on the tasks that we investigated. We also investigated an additional set of baselines that combined our pretrained slot attention module with various reasoning architectures, including GAMR [43], ESBN [51], Relation Net (RN) [35], Transformer [45], Interaction Network (IN) [48], and CoRelNet [22]. These baselines benefited from the same degree of pre-training and object-centric processing as our proposed model, but employed different reasoning mechanisms. Finally, we investigated a self-supervised baseline (a Masked Autoencoder (MAE) [16]) to assess the extent to which a generative objective might encourage systematic generalization (see Tables S13-S14 for results). All baselines were evaluated using multi-object displays. See Supplementary Section S3 for more details on baselines.

## 4.3 Experimental Details

### 4.3.1 Pre-training slot attention

We pre-trained slot attention using a slot-based autoencoder framework. For ART, we generated a dataset of random multi-object displays (see Figure S2), utilizing a publicly available repository of unicode character images[2]. We eliminated any characters that were identical to, or closely resembled, the 100 characters present in the ART dataset, resulting in a training set of 180k images, and a validation set of 20k images. For SVRT, we pre-trained two versions of slot attention on all tasks, one using 500 and the other using 1000 training examples from each of the 23 tasks. For CLEVR-ART, we generated a dataset of randomly placed 3D shapes from CLEVR, and pre-trained slot attention using a training set of 90k images, and validation set of 10k images.

To pre-train slot attention, we used a simple reconstruction objective and a slot-based decoder. Specifically, we used a spatial broadcast decoder [47], which generates both a reconstructed image and a mask for each slot. We then generated a combined reconstruction, normalizing the masks across slots using a softmax, and using the normalized masks to compute a weighted average of the slot-specific reconstructions.

After pre-training, the slot attention parameters were frozen for training on the downstream reasoning tasks. For ART, we found that despite not having been trained on any of the objects in the dataset, slot attention was nevertheless able to generate focused attention maps that corresponded closely to the location of the objects (see Figures S6-S9).

### 4.3.2 Training details and hyperparameters

Images were resized to $128 \times 128$, and pixels were normalized to the range $[0, 1]$. For SVRT, we converted the images to grayscale, and also applied random online augmentations during training, including horizontal and vertical flips. Tables S2 and S3 describes the hyperparameters for the convolutional encoder and spatial broadcast decoder respectively. For slot attention, we used $K = 6$ ($K = 7$ for CLEVR-ART) slots, $T = 3$ attention iterations per image, and a dimensionality of $D = 64$. Training details are described in Table S4.

Table S5 gives the hyperparameters for the transformer. To solve the SVRT tasks, and the same/different ART task, we used a CLS token (analogous to CLS token in [9]) together with a sigmoidal output layer to generate a binary label, and trained the model with binary cross-entropy

---

[2]https://github.com/bbvanexttechnologies/unicode-images-database

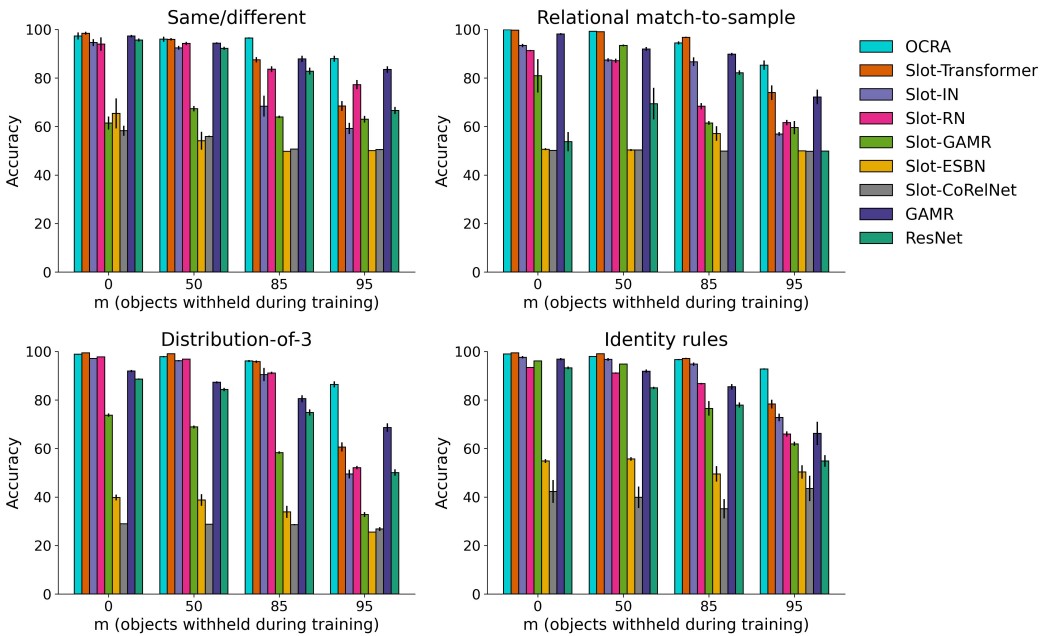

Figure 3: **ART results.** Results on the Abstract Reasoning Tasks dataset [51]. OCRA outperformed all baselines for all tasks, with especially strong performance in the most difficult generalization regime ($m = 95$). Results reflect an average across 10 runs for each model $\pm$ standard error. The X axis denotes the number of objects (out of 100) withheld during training, with no objects withheld in the leftmost condition ($m = 0$), and almost all objects withheld in the rightmost condition ($m = 95$).

loss. To solve the RMTS, distribution-of-3, and identity rules tasks, we inserted each candidate answer choice into the problem, and used a CLS token together with a linear output layer to generate a score. The scores for all answer choices were passed through a softmax and the model was trained with cross-entropy loss. For ART and CLEVR-ART, we used a default learning rate of $8e - 5$, and a batch size of 16. The number of training epochs is displayed for each ART task in Table S6, and for each CLEVR-ART task in Table S10. The training details for some baseline models were modified as discussed in Section S3, and detailed in Tables S7-S9. For SVRT, we used a learning rate of $4e - 5$, a batch size of 32, and trained for 2000 epochs on each task. We used the ADAM optimizer [23], and implementation was done using the Pytorch library [31].

### 4.3.3 Model selection

After pre-training slot attention for a fixed number of epochs (number of epochs for each dataset listed in Table S4), we selected the version of the model from the epoch with the lowest validation loss (on the unsupervised pre-training task) to use in the visual reasoning tasks. We used a validation set of 20k images for ART (again excluding the objects used in the ART tasks), 4k images per task for SVRT, and 10k images for CLEVR-ART.

When training on the primary reasoning tasks, for SVRT, we trained for 2000 epochs on each task, and selected the version of the model from the epoch with the highest accuracy on the validation set. This is consistent with the approach taken by Vaishnav and Serre [43], who used validation accuracy as a criterion for early stopping. For ART and CLEVR-ART, we did not perform any model (or epoch) selection when training on the primary reasoning tasks.

## 5 Results

Figure 3 shows the results on the ART dataset (all results are presented in tabular form in Tables S11-S17). OCRA achieved state-of-the-art results across all tasks and generalization regimes, outperforming both the previous state-of-the-art model (GAMR), and other strong baselines such as

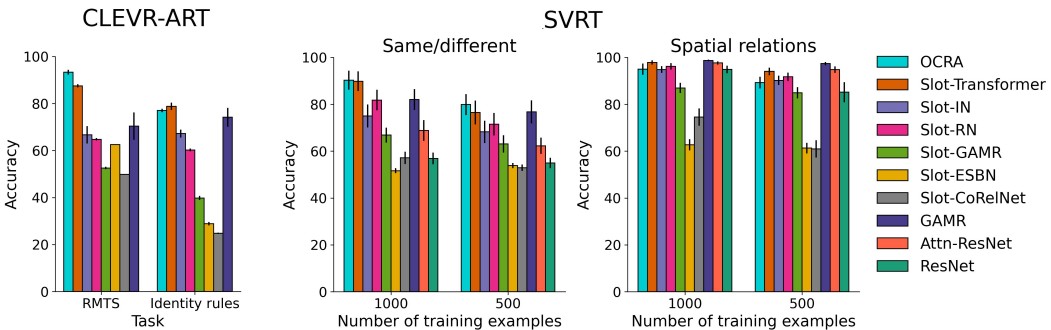

Figure 4: **CLEVR-ART.** Results for systematic generalization on two CLEVR-ART tasks. Results reflect test accuracy averaged over 5 trained networks ± standard error. **SVRT.** Results on the Synthetic Visual Reasoning Test (SVRT) [13]. OCRA and GAMR showed comparable overall performance, with OCRA peforming marginally better on problems involving same / different relations, and GAMR performing marginally better on problems involving spatial relations.

the slot-transformer. This disparity was especially striking in the most difficult generalization regime ($m = 95$), in which models were trained on only 5 objects, and tested on a heldout set of 95 objects. Importantly, no component of OCRA (including both the slot attention and relational reasoning components) had been exposed to these heldout objects. OCRA outperformed the next-best baselines in this regime (GAMR and slot-transformer) by as much as 25%, demonstrating a significant degree of systematic generalization. We also analyzed OCRA's learned relational embeddings, finding that they captured relational information even in this out-of-distribution setting (Figure S5).

Many of the baseline models that we evaluated did not contain a relational bottleneck (Transformer, IN, RN, GAMR, ResNet), limiting their ability to generalize relational patterns to novel objects (i.e., higher values of $m$). The comparison with these models thus demonstrates the importance of the relational bottleneck, which enables systematic generalization of learned relational patterns. The ESBN and CoRelNet architectures *did* include a relational bottleneck, but they were not designed with multi-object inputs in mind, and therefore performed poorly on these tasks in all regimes (even when combined with our slot attention module). This is due primarily to the random permutation of slots in slot attention, which motivated our positional embedding variable-binding scheme (allowing OCRA to keep track of which relations correspond to which pairs of objects).

On SVRT, OCRA showed comparable overall performance with GAMR, displaying marginally better performance on tasks characterized by same/different relations, while GAMR performed marginally better on tasks involving spatial relations (Figure 4, right). OCRA also outperformed GAMR on CLEVR-ART (Figure 4, left), demonstrating that its capacity for systematic generalization can be extended to problems involving more realistic visual inputs. Overall, the results on ART, SVRT, and CLEVR-ART demonstrate that OCRA is capable of solving a diverse range of relational reasoning problems from complex multi-object displays.

## 5.1 Ablation Study

We performed an ablation study targeting each of OCRA's major components, focusing on the most difficult regime of the relational match-to-sample and identity rules tasks (Table 1). To evaluate the importance of OCRA's capacity for object-centric processing, we tested a version of the model without slot attention ('w/o Slot Attention'), in which the visual feature map was instead divided into a $4 \times 4$ grid, treating each image patch as though it were an 'object'. We also tested a version in which slot attention was only trained end-to-end on the visual reasoning tasks ('w/o Pre-training'), to test the importance of prior exposure to a diverse range of multi-object displays. To evaluate the importance of OCRA's relational embedding method, we tested five different ablation models. In one model ('w/o Relational Embeddings'), slot embeddings were passed directly to the transformer. In another model ('w/o Factorized Representation'), we did not compute separate feature and position embeddings for each object, and instead applied the relational operator directly to the slot embeddings. In a third model ('w/o Relational Bottleneck'), relational embeddings were computed simply by applying an MLP to each pair of slot embeddings (allowing the model to overfit to the perceptual

Table 1: Ablation study on $m = 95$ generalization regime for two ART tasks. Results reflect test accuracy averaged over 10 trained networks $\pm$ standard error.

|  | RMTS | ID |
|---|---|---|
| OCRA (full model) | **85.31$\pm$2.0** | **92.80$\pm$0.3** |
| w/o Slot Attention | 56.58$\pm$3.1 | 84.42$\pm$1.1 |
| w/o Pre-training | 58.69$\pm$3.6 | 53.48$\pm$4.8 |
| w/o Relational Embeddings | 74.79$\pm$1.8 | 81.21$\pm$1.1 |
| w/o Factorized Representation | 50.26$\pm$0.2 | 75.40$\pm$0.6 |
| w/o Relational Bottleneck | 63.62$\pm$1.0 | 76.56$\pm$1.8 |
| w/o Inner Product | 50.70$\pm$0.4 | 48.07$\pm$2.6 |
| Replace Inner Product w/ Outer Product | 62.84$\pm$1.6 | 69.58$\pm$1.1 |
| w/o Transformer | 51.29$\pm$0.2 | 44.18$\pm$0.6 |

details of these embeddings), thus removing OCRA's inductive bias for relational abstraction. In a fourth model ('w/o Inner Product') the dot product in the relational operator (Equation 3) was replaced with a learned bottleneck, in which each pair of slot embeddings was passed to a learned linear layer with a one-dimensional output. This model tested the extent to which the inherently relational properties of the dot product are important for relational abstraction (as opposed to resulting merely from compression to a single dimension). In a fifth model ('Replace Inner Product w/ Outer Product'), the dot product was replaced with an outer product (resulting in a $D \times D$ matrix, which was then flattened and projected via a learned linear layer to an embedding of size $D$). Finally, to evaluate the importance of OCRA's capacity to process higher-order relations, instead of passing relational embeddings to a transformer, we instead summed all relation embeddings elementwise and passed the resulting vector to an MLP, as is done in the Relation Net ('w/o Transformer'). All ablation models performed significantly worse than the complete version of OCRA, demonstrating that all three elements in our model – object-centric processing, relational abstraction, and higher-order relations – play an essential role in enabling systematic generalization.

## 6  Limitations and Future Directions

In the present work, we have presented a model that integrates object-centric visual processing mechanisms (providing the ability to operate over complex multi-object visual inputs) with a relational bottleneck (providing a strong inductive bias for learning relational abstractions that enable human-like systematic generalization of learned abstract rules). Though this is a promising step forward, and a clear advance relative to previous models of abstract visual reasoning, it is likely that scaling the present approach to real-world settings will pose additional challenges. First, real-world images are especially challenging for object-centric methods due to a relative lack of consistent segmentation cues. However, there has recently been significant progress in this direction [36], in particular by utilizing representations from large-scale self-supervised methods [54, 7], and it should be possible to integrate these advances with our proposed framework. Second, the current approach assumes a fixed number of slot representations, which may not be ideal for modeling real-world scenes with a highly variable number of objects [55]. Though we did not find that this was an issue in the present work, in future work it would be desirable to develop a method for dynamically modifying the number of slots. Third, OCRA's relational operator is applied to all possible pairwise object comparisons, an approach that may not be scalable to scenes that contain many objects. In future work, this may be addressed by replacing the transformer component of our model with architectures that are better designed for high-dimensional inputs [18]. Finally, it is unclear how our proposed approach may fare in settings that involve more complex real-world relations, and settings that require the discrimination of multiple relations at once. It may be beneficial in future work to investigate a 'multi-head' version of our proposed model (analogous to multi-head attention), in which multiple distinct relations are processed in parallel. A major challenge for future work is to develop models that can match the human capacity for structured visual reasoning in the context of complex, real-world visual inputs.

# 7 Acknowledgements

Shanka Subhra Mondal was supported by Office of Naval Research grant N00014-22-1-2002 during the duration of this work. We would like to thank Princeton Research Computing, especially William G. Wischer and Josko Plazonic, for their help with scheduling training jobs on the Princeton University Della cluster.

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
