# Supplementary Material

## S1    Code and Data Availability

## S2    Datasets

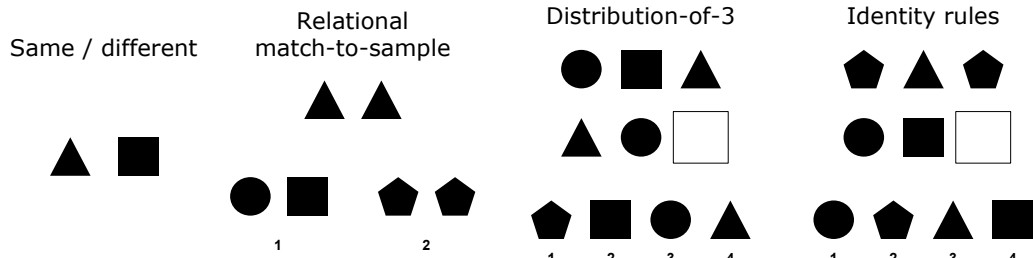

Figure S1: **Abstract Reasoning Tasks (ART). Same/different:** Two objects are presented, and the task is to say whether they are the same or different. **Relational match-to-sample:** A source pair of objects is presented that either instantiates a 'same' or 'different' relation, and the task is to select the pair of target objects (out of two pairs) that instantiates the same relation. Problems were presented in a $2 \times 2$ array format, with the source pair presented in the top row, and a target pair presented in the bottom row (separate images for each target pair, see Figure S7). **Distribution-of-3:** A set of three objects is presented in the first row, and an incomplete set is presented in the second row. The task is to select the missing object from a set of four choices. Problems were presented in a $2 \times 3$ array format, with one of the answer choices inserted into the bottom right cell (separate images for each answer choice, see Figure S8). **Identity rules:** An abstract pattern is instantiated in the first row (ABA, ABB, or AAA), and the task is to select the choice that would result in the same relation being instantiated in the second row. Problems were presented in the same format as the distribution-of-3 task (Figure S9).

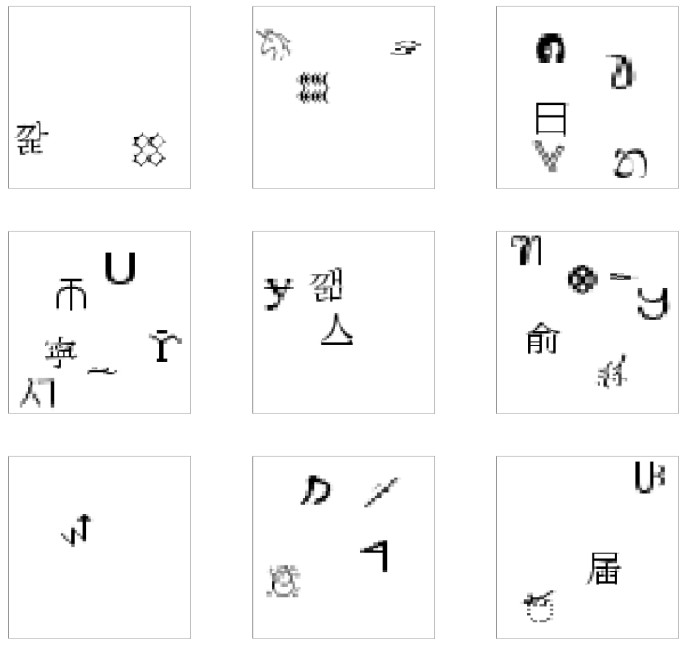

Figure S2: A few examples of the multi-object inputs used to pre-train slot attention for ART.

Table S1: Number of training and test samples for ART dataset.

| TASK | | $m = 95$ | $m = 85$ | $m = 50$ | $m = 0$ |
|------|------|------|------|------|------|
| SD | TRAINING | 40 | 420 | 4900 | 18810 |
| | TEST | 10000 | 10000 | 4900 | 990 |
| RMTS | TRAINING | 480 | 10000 | 10000 | 10000 |
| DIST3 | TRAINING | 360 | 10000 | 10000 | 10000 |
| ID | TRAINING | 8640 | 10000 | 10000 | 10000 |
| | TEST | 10000 | 10000 | 10000 | 10000 |

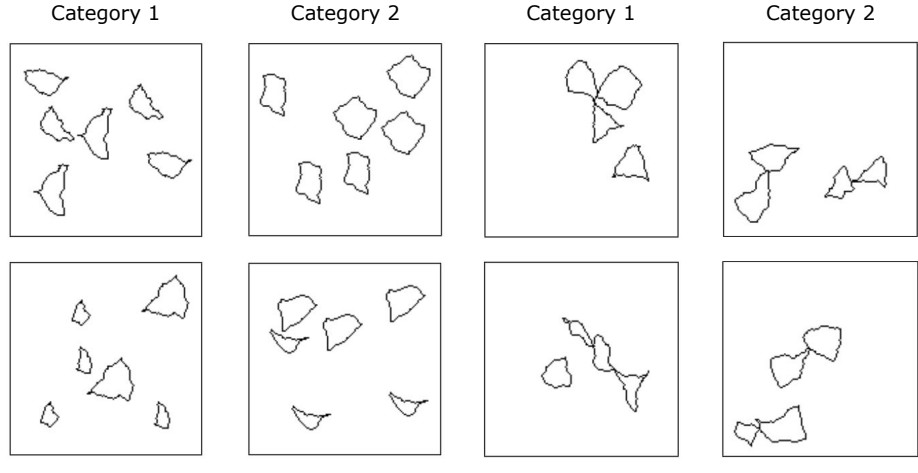

(a) Example same/different (SD) task.     (b) Example spatial relation (SR) task.

Figure S3: **Example tasks from Synthetic Visual Reasoning Test (SVRT). (a)** Example same/different (SD) task. Panels depict two examples from each of two categories for task 7. In category 1, there are always three sets of two identical objects. In category 2, there are always two sets of three identical objects. **(b)** Example spatial relation (SR) task. Panels depict two examples from each of two categories for task 3. In category 1, three out of four objects are in contact while the fourth object is positioned separately. In category 2, there are two sets of two objects in contact.

Relational match-to-sample

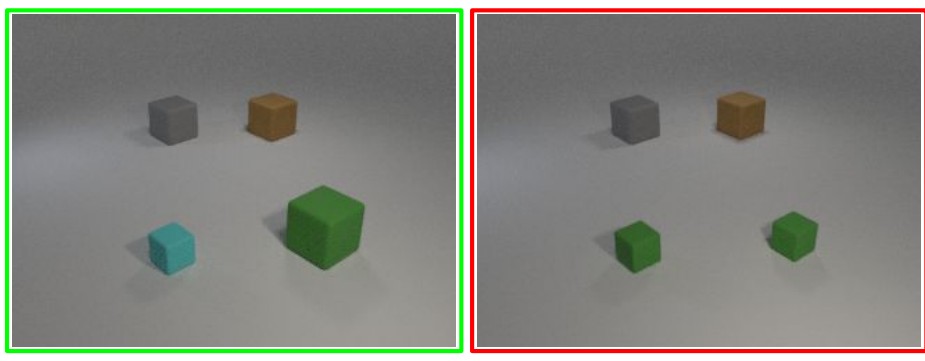

Identity rules

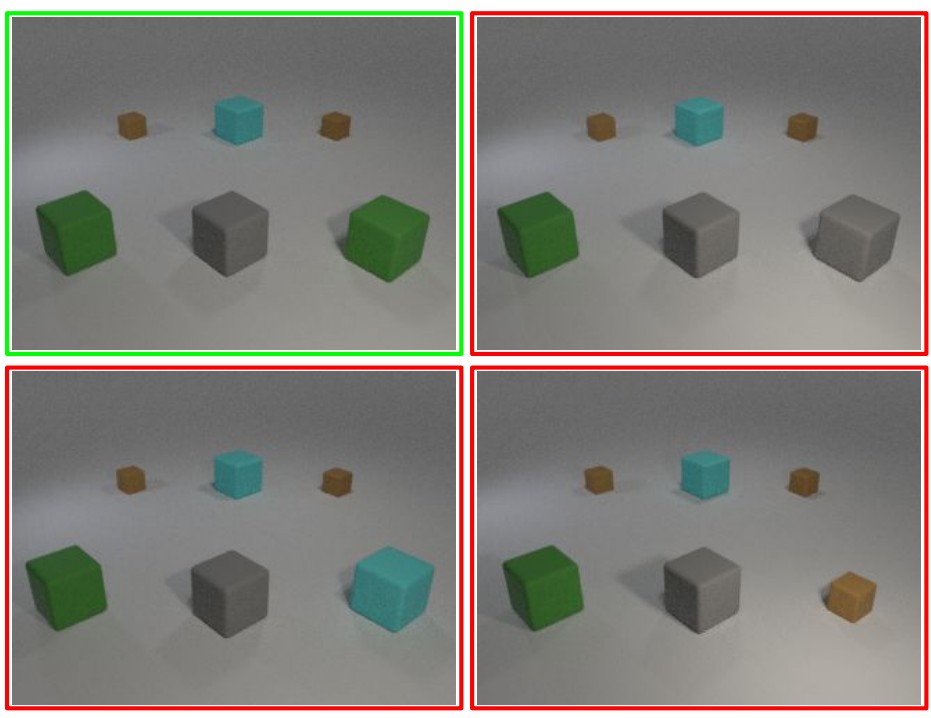

Figure S4: **CLEVR-ART. Relational match-to-sample:** Example problem involving 'different' relation. Correct answer choice (left image) involves 'different' relation for both source pair (back row of objects) and target pair (front row of objects). Incorrect answer choice (right image) involves 'same' relation in target pair. **Identity rules:** Example problem involving ABA rule. Correct answer choice (top left image) involves ABA rule in both back row and front row of objects. Front right object in the other three images (incorrect choices) violates this rule.

## S3 Baselines

### S3.1 GAMR

For the ART and SVRT datasets, we compared to results for the GAMR baseline as originally reported in [43]. For CLEVR-ART, we used the implementation of GAMR provided at https://openreview.net/forum?id=iLMgk2IGNyv (Supplementary Material).

### S3.2 ResNet and Attn-ResNet

For ART and SVRT, we compared to results from ResNet50 [17] as originally reported in [43]. For SVRT, we also compared to results from a version of ResNet that employed self-attention (Attn-ResNet), as reported in [43]. Attn-ResNet was similar to ResNet50, except that it included a transformer-style self-attention layer between ResNet blocks (see [43] for details).

### S3.3 Slot-based reasoning baselines

We investigated a number of baseline models that combined our pre-trained slot attention module with alternative reasoning architectures. The slot attention module was the same as used in OCRA. The details of the reasoning architectures are described in the following sections. To train these baselines, we used the same settings as used for OCRA (learning rate, batch size, and number of training epochs), unless otherwise specified.

#### S3.3.1 Slot-GAMR

To combine slot attention with GAMR, we replaced $z_{img}$ in the original model (originally a flattened feature map from a convolutional encoder) with the concatenated slot embeddings $slots \in \mathbb{R}^{K \times D}$. We trained slot-GAMR with a batch size of 32. The learning rate and number of training epochs for all ART tasks is given in Tables S8 and S7 respectively. These training details were based on those used in the original work [43], except that we used a different learning rate and trained the model for longer in some cases in order to achieve convergence on the training set.

#### S3.3.2 Slot-ESBN

To combine slot attention with the Emergent Symbol Binding Network (ESBN), the slot embeddings were passed sequentially to the ESBN, replacing $z_{t=1...T}$ in the original architecture (one slot embedding per timestep). The architectural details for ESBN were the same as reported in [51], except that the output layer from the LSTM controller was changed based on the task (as described in Section 4.3.2). We used the implementation provided at https://github.com/taylorwwebb/emergent_symbols. We trained slot-ESBN with a batch size of 32 and a learning rate of $5e-4$. The number of training epochs for all ART tasks is given in Table S9. These training details were based on those used in the original work [51], except that we trained the model for longer in some cases.

#### S3.3.3 Slot-RN

For the slot-RN baseline, we first passed each pair of slot embeddings through a shared MLP (referred to as $g_\theta$ in the RN framework [35]), with a hidden layer of size 512, an output layer of size 256, and ReLU nonlinearities in both layers. The outputs of this MLP were then summed elementwise and passed to a second MLP ($f_\phi$), with a hidden layer of size 256 and ReLU nonlinearities, and a single output unit (the nonlinearity applied to this output depended on the task, as described in Section 4.3.2).

#### S3.3.4 Slot-Transformer

For the slot-transformer baseline, we passed the slot embeddings directly to the transformer used in OCRA. Note that this is identical to the ablation model referred to as '- Relation Embeddings' in Table 1.

### S3.3.5 Slot-IN

For the Slot-IN baseline, the slot embeddings were passed to an Interaction Net (IN), with hyperparameters identical to those described in [48] (the 'Interaction Net' described in Section 8.2 of that work). Slot embeddings were updated for $T = 6$ iterations, then summed elementwise and passed through a final MLP with a hidden layer of size 64 and ReLU nonlinearities, and a single output unit (with the nonlinearity depending on the task, as described in Section 4.3.2).

### S3.3.6 Slot-CoRelNet

For the Slot-CoRelNet baseline, we computed the matrix of all pairwise dot products between slot embeddings, applied a softmax function across the rows of this matrix, flattened the matrix and passed it to an MLP with the same hyperparameters described in [22].

### S3.4 MAE

We applied the Masked Autoencoder (MAE) model [16] on the identity rules and distribution-of-3 ART tasks by masking out the final object in each problem (in the bottom right cell of the input), and training the model to fill in this patch. To select from the set of multiple choices, we then compared the model's generated output with the four answer choices, and selected the choice with lowest mean-squared error. We used the same hyperparameters described in [16] for the MAE architecture. We trained the model for 400 epochs with a batch size of 64, using a learning rate of $2.5e^{-4}$ with warmup for 40 epochs followed by cosine learning rate decay. We did not test this model on the same/different or RMTS tasks, as it was not clear how to formulate these tasks in a generative manner.

## S4 Training details and hyperparameters

Before computing relational embeddings, we applied temporal context normalization (TCN) [49] to both the the feature embeddings and the position embeddings. TCN normalizes representations across the temporal dimension, and has been shown to improve generalization in relational reasoning tasks. When applying the relational operator $\phi$, we also softmax-normalized the dot products for all pairwise comparisons.

As shown in Table S4, we pretrained slot attention for SVRT using 500 training examples for each of the 23 tasks with a learning rate of $4e - 4$ for the first 1350 epochs, followed by using a learning rate of $8e - 5$ for the next 4620 epochs. When using 1000 training examples for each of the 23 tasks we first pretrained slot attention with a learning rate of $4e - 4$ for the first 1500 epochs, followed by using a learning rate of $8e - 5$ for the next 3510 epochs.

Table S2: CNN encoder hyperparameters.

| TYPE | CHANNELS | ACTIVATION | KERNEL SIZE | STRIDE | PADDING |
|---|---|---|---|---|---|
| 2D CONV | 64 | RELU | $5 \times 5$ | 1 | 2 |
| 2D CONV | 64 | RELU | $5 \times 5$ | 1 | 2 |
| 2D CONV | 64 | RELU | $5 \times 5$ | 1 | 2 |
| 2D CONV | 64 | RELU | $5 \times 5$ | 1 | 2 |
| POSITION EMBEDDING | - | - | - | - | - |
| FLATTEN | - | - | - | - | - |
| LAYER NORM | - | - | - | - | - |
| 1D CONV | 64 | RELU | 1 | 1 | 0 |
| 1D CONV | 64 | - | 1 | 1 | 0 |

Table S3: Slot decoder hyperparameters.

| TYPE | CHANNELS | ACTIVATION | KERNEL SIZE | STRIDE | PADDING |
|------|----------|------------|-------------|--------|---------|
| SPATIAL BROADCAST | - | - | - | - | - |
| POSITION EMBEDDING | - | - | - | - | - |
| 2D CONV | 64 | RELU | $5 \times 5$ | 1 | 2 |
| 2D CONV | 64 | RELU | $5 \times 5$ | 1 | 2 |
| 2D CONV | 64 | RELU | $5 \times 5$ | 1 | 2 |
| 2D CONV | 64 | RELU | $5 \times 5$ | 1 | 2 |
| 2D CONV | 64 | RELU | $5 \times 5$ | 1 | 2 |
| 2D CONV | 2 | - | $3 \times 3$ | 1 | 1 |

Table S4: Slot attention pre-training details for all datasets.

| | ART | SVRT Dataset Size = 0.5k | SVRT Dataset Size = 1k | CLEVR-ART |
|------|-----|--------------------------|------------------------|-----------|
| Batch size | 32 | 64 | 64 | 32 |
| Learning rate | $8e-5$ | $4e-4, 8e-5$ | $4e-4, 8e-5$ | $8e-5$ |
| LR warmup steps | 150k | 9k | 18k | 90k |
| Epochs | 750 | 1350, 4620 | 1500, 3510 | 1000 |

Table S5: Hyperparameters for Transformer reasoning module. $H$ is the number of heads, $L$ is the number of layers, $D_{head}$ is the dimensionality of each head, and $D_{MLP}$ is the dimensionality of the MLP hidden layer.

| | ART | SVRT | CLEVR-ART |
|------|-----|------|-----------|
| $H$ | 8 | 8 | 8 |
| $L$ | 6 | 24 | 24 |
| $D_{head}$ | 64 | 64 | 64 |
| $D_{MLP}$ | 512 | 512 | 512 |
| DROPOUT | 0.1 | 0 | 0 |

Table S6: Default number of training epochs for ART tasks.

| TASK | $m=0$ | $m=50$ | $m=85$ | $m=95$ |
|------|-------|--------|--------|--------|
| SD | 100 | 100 | 100 | 600 |
| RMTS | 100 | 100 | 100 | 400 |
| DIST3 | 100 | 100 | 100 | 400 |
| ID | 100 | 100 | 100 | 100 |

Table S7: Number of training epochs for slot-GAMR baseline on ART tasks.

| TASK | $m=0$ | $m=50$ | $m=85$ | $m=95$ |
|------|-------|--------|--------|--------|
| SD | 50 | 50 | 100 | 200 |
| RMTS | 100 | 50 | 50 | 300 |
| DIST3 | 100 | 150 | 150 | 300 |
| ID | 100 | 100 | 50 | 100 |

Table S8: Learning rate for slot-GAMR baseline on ART tasks.

| TASK | $m = 0$ | $m = 50$ | $m = 85$ | $m = 95$ |
|------|---------|----------|----------|----------|
| SD | $1e-4$ | $5e-4$ | $5e-4$ | $1e-3$ |
| RMTS | $5e-4$ | $1e-4$ | $5e-4$ | $5e-4$ |
| DIST3 | $5e-4$ | $1e-4$ | $5e-5$ | $5e-4$ |
| ID | $5e-4$ | $5e-4$ | $5e-4$ | $5e-4$ |

Table S9: Number of training epochs for slot-ESBN baseline on ART tasks.

| TASK | $m = 0$ | $m = 50$ | $m = 85$ | $m = 95$ |
|------|---------|----------|----------|----------|
| SD | 150 | 150 | 150 | 200 |
| RMTS | 150 | 150 | 150 | 200 |
| DIST3 | 150 | 150 | 150 | 200 |
| ID | 150 | 150 | 150 | 150 |

Table S10: Number of training epochs for CLEVR-ART tasks. OCRA was trained for longer on the identity rules (ID) task in order to achieve convergence on the training set. 50 epochs was sufficient to achieve convergence on both tasks for GAMR, and on RMTS for OCRA.

| TASK | OCRA | GAMR |
|------|------|------|
| RMTS | 50 | 50 |
| ID | 200 | 50 |

## S5 Results

Table S11: Results for ART same/different task.

| | $m = 0$ | $m = 50$ | $m = 85$ | $m = 95$ |
|------|---------|----------|----------|----------|
| OCRA (OURS) | **97.31±1.5** | **95.98±1.1** | **96.48±0.3** | **87.95±1.3** |
| SLOT-TRANSFORMER | **98.47±0.6** | **95.95±0.5** | 87.53±1.1 | 68.46±2.0 |
| SLOT-IN | 94.59±1.4 | 92.43±0.8 | 68.4±4.3 | 59.23±2.3 |
| SLOT-RN | 93.97±2.8 | 94.23±0.7 | 83.66±1.2 | 77.26±1.9 |
| SLOT-GAMR | 61.43±2.7 | 67.35±1.1 | 63.97±0.5 | 62.98±1.4 |
| SLOT-ESBN | 65.42±6.2 | 54.1±3.7 | 49.79±0.1 | 50.02±0.2 |
| SLOT-CORELNET | 58.24±2.1 | 55.91±0.3 | 50.70±0.2 | 50.50±0.2 |
| GAMR | **97.28±0.5** | **94.4±0.3** | 87.88±1.3 | 83.49±1.4 |
| RESNET | 95.65±0.7 | 92.23±0.6 | 82.83±1.4 | 66.6±1.5 |

Table S12: Results for ART relational-match-to-sample task.

| | $m = 0$ | $m = 50$ | $m = 85$ | $m = 95$ |
|------|---------|----------|----------|----------|
| OCRA (OURS) | **99.91±0.0** | **99.25±0.1** | 94.43±0.7 | **85.31±2.0** |
| SLOT-TRANSFORMER | **99.73±0.0** | **99.12±0.1** | **96.77±0.3** | 73.99±3.0 |
| SLOT-IN | 93.42±0.7 | 87.42±0.7 | 86.71±1.8 | 56.93±0.8 |
| SLOT-RN | 91.31±0.3 | 87.1±0.9 | 68.4±1.3 | 61.62±1.1 |
| SLOT-GAMR | 80.92±6.9 | 93.43±0.4 | 61.47±0.8 | 59.55±2.7 |
| SLOT-ESBN | 50.63±0.4 | 50.3±0.4 | 57.12±3.0 | 49.99±0.2 |
| SLOT-CORELNET | 50.18±0.1 | 50.30±0.2 | 49.84±0.2 | 49.82±0.2 |
| GAMR | **98.12±0.4** | 91.98±0.8 | 89.81±0.5 | 72.2±3.0 |
| RESNET | 53.77±4.0 | 69.43±6.5 | 82.18±1.0 | 49.89±0.2 |

Table S13: Results for ART distribution-of-3 task.

|  | $m = 0$ | $m = 50$ | $m = 85$ | $m = 95$ |
|---|---|---|---|---|
| OCRA (OURS) | **98.86±0.2** | **97.87±0.3** | **96.09±0.4** | **86.42±1.3** |
| SLOT-TRANSFORMER | **99.49±0.0** | **99.08±0.1** | 95.82±0.5 | 60.61±1.9 |
| SLOT-IN | 97.12±0.2 | 96.22±0.3 | 90.51±2.7 | 49.48±1.8 |
| SLOT-RN | 97.8±0.1 | 96.82±0.1 | 91.14±0.5 | 52.1±0.7 |
| SLOT-GAMR | 73.74±0.7 | 68.91±0.6 | 58.3±0.5 | 32.77±1.0 |
| SLOT-ESBN | 39.79±1.2 | 38.8±2.4 | 33.86±2.5 | 25.56±0.1 |
| SLOT-CORELNET | 28.98±0.1 | 28.78±0.1 | 28.58±0.3 | 26.80±0.8 |
| GAMR | 91.98±0.5 | 87.3±0.4 | 80.57±1.4 | 68.62±1.8 |
| RESNET | 88.61±0.3 | 84.32±0.7 | 74.82±1.3 | 50.07±1.3 |
| MAE | **99.99±0.0** | 56.47±1.1 | 40.90±1.2 | 28.85±0.9 |

Table S14: Results for ART identity rules task.

|  | $m = 0$ | $m = 50$ | $m = 85$ | $m = 95$ |
|---|---|---|---|---|
| OCRA (OURS) | **99.01±0.0** | **98.01±0.1** | **96.67±0.2** | **92.8±0.3** |
| SLOT-TRANSFORMER | **99.44±0.0** | **99.06±0.1** | **97.15±0.3** | 78.32±1.8 |
| SLOT-IN | 97.62±0.5 | 96.72±0.6 | 94.78±0.7 | 72.82±1.6 |
| SLOT-RN | 93.44±0.2 | 91.1±0.3 | 86.73±0.3 | 65.96±1.1 |
| SLOT-GAMR | 96.11±0.1 | 94.8±0.2 | 76.52±3.0 | 61.92±0.9 |
| SLOT-ESBN | 54.83±0.8 | 55.7±0.7 | 49.52±3.2 | 50.33±2.8 |
| SLOT-CORELNET | 42.28±4.7 | 39.84±4.4 | 35.19±4.0 | 43.50±5.2 |
| GAMR | 96.89±0.5 | 91.9±0.7 | 85.44±1.2 | 66.23±4.8 |
| RESNET | 93.27±0.6 | 84.97±0.6 | 77.94±1.1 | 54.84±2.4 |
| MAE | 66.55±0.2 | 44.96±0.6 | 38.39±0.9 | 31.56±1.0 |

Table S15: Results for systematic generalization on two CLEVR-ART tasks. Results reflect test accuracy averaged over 5 trained networks (± standard error).

|  | RMTS | ID |
|---|---|---|
| GAMR | 70.40±5.8 | 74.15±4.0 |
| Slot-Transformer | 87.54±0.7 | **78.81±1.6** |
| Slot-IN | 66.72±3.7 | 67.22±1.7 |
| Slot-RN | 64.79±0.5 | 60.27±0.6 |
| Slot-GAMR | 52.56±0.5 | 39.83±0.9 |
| Slot-ESBN | 62.53±0.1 | 28.87±0.7 |
| Slot-CoRelNet | 49.87±0.2 | 24.80±0.3 |
| OCRA | **93.34±1.0** | **77.06±0.7** |

Table S16: Results for SVRT same/different tasks.

|  | DATASET SIZE =1K | DATASET SIZE =0.5K |
|---|---|---|
| OCRA (OURS) | **90.30±4.1** | **79.89±4.5** |
| SLOT-TRANSFORMER | **89.85±4.2** | **76.54±5.1** |
| SLOT-IN | 74.99±4.9 | 68.23±4.8 |
| SLOT-RN | 81.79±4.4 | 71.48±4.8 |
| SLOT-GAMR | 66.87±3.2 | 63.06±3.7 |
| SLOT-ESBN | 51.67±1.1 | 53.83±1.1 |
| SLOT-CORELNET | 57.13±2.6 | 52.95±1.4 |
| GAMR | 82.05±4.4 | **76.80±4.9** |
| ATTN-RESNET | 68.83±4.4 | 62.30±3.5 |
| RESNET | 56.88±2.5 | 54.97±2.2 |

Table S17: Results for SVRT spatial-relations tasks.

|  | DATASET SIZE =1K | DATASET SIZE =0.5K |
|---|---|---|
| OCRA (OURS) | 95.02 ± 2.4 | 89.25 ± 2.5 |
| SLOT-TRANSFORMER | **97.86±0.9** | 94.06±1.6 |
| SLOT-IN | 94.86±1.4 | 90.23±2.0 |
| SLOT-RN | 96.20±1.4 | 91.73±1.8 |
| SLOT-GAMR | 86.99±2.2 | 84.90±2.4 |
| SLOT-ESBN | 62.69±2.4 | 61.30±2.3 |
| SLOT-CORELNET | 74.59±3.7 | 60.95±3.7 |
| GAMR | **98.74±0.3** | **97.40±0.7** |
| ATTN-RESNET | 97.66±0.7 | 94.80±1.4 |
| RESNET | 94.87±1.6 | 85.18±4.3 |

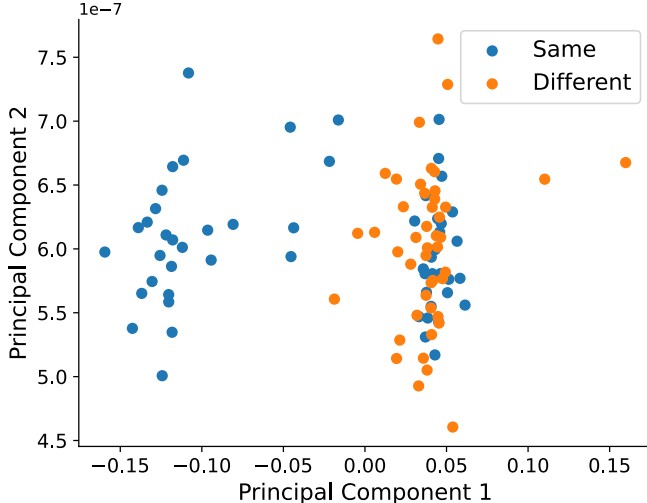

Figure S5: **Visualization of learned relation embeddings.** OCRA's relation embeddings (projected on to top 2 principal components) for test set in $m = 95$ regime of same/different ART task. Despite completely novel objects, relation embeddings display separation based on relational category.

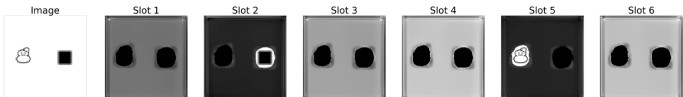

Figure S6: Slot-specific attention maps applied to input image for same/different task.

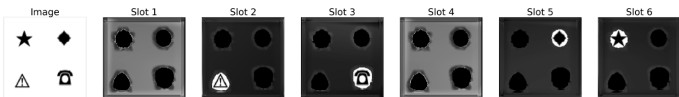

Figure S7: Slot-specific attention maps applied to input image for relational match-to-sample task.

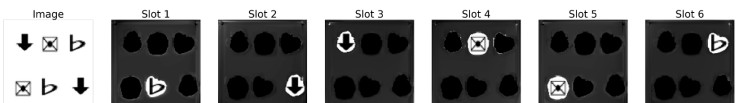

Figure S8: Slot-specific attention maps applied to input image for distribution-of-three task.

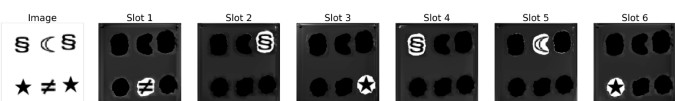

Figure S9: Slot-specific attention maps applied to input image for identity rules task.