# OpenReview forum: "Systematic Visual Reasoning through Object-Centric Relational Abstraction"
_NeurIPS.cc/2023/Conference — NeurIPS 2023 poster_

### Official Review · Reviewer_6gd3 · 2023-07-04

**Soundness:** 3 good
**Presentation:** 3 good
**Contribution:** 2 fair
**Rating:** 6
**Confidence:** 3

**Summary:**

The paper proposes to combine an object-centric representation model (Slot Attention) with a relational reasoning module to create the Object-Centric Relational Abstraction (OCRA) model. For this model, Slot Attention is first pre-trained in an unsupervised way to represent objects separately. Then, the relational reasoning module, consisting of a "relational bottleneck" and a transformer model, is applied to these object-centric representations. This reasoning model is trained in a supervised way on a subset of possible objects. The experiments on three different datasets show that this approach generalizes better to previously unseen objects than comparable baselines.


**Strengths:**

The paper is written well and easy to follow. The introduction provides a strong motivation for this line of research. The proposed approach and conducted experiments are described clearly and with sufficient detail, such that it should be possible to reproduce them.

The proposed approach seems to be novel and achieves strong generalization performance on previously unseen objects.

**Weaknesses:**

In my eyes, the main weakness of this paper is that its contribution relative to existing work is not entirely clear.

1) Most importantly, the paper positions itself as proposing an approach with a stronger inductive bias toward relational abstraction, which ultimately leads to better generalization performance (for example, introduction (l.46) and related work section (l. 135, l. 170). However, the experiments do not compare against many existing approaches. Without such a comparison, it remains unclear whether the existing approaches actually suffer from the problem that the proposed approach claims to solve. Thus, ideally, the paper should provide such a comparison, or alternatively adjust the positioning of the proposed approach relative to existing work.

2) Besides this, it remains somewhat unclear which parts of the proposed approach are building on existing work (such as Slot Attention) and which parts are new. For example, do equations (1) and (2) and sections 2.2 and 2.3 describe novel architectural components or have they been used in a similar fashion before?

---

3) The related work section fails to cite many relevant papers. Object-centric representation learning has seen growing interest in recent years, with many papers proposing various solutions. I would recommend taking a look at the related works section of [1], which provides a comprehensive overview of papers that are relevant to this work. Besides this, [2] describes a recent model that combines object-centric representation learning with relational reasoning capabilities and [3] describes generalization properties of Slot Attention to previously unseen objects.


4) The experimental results could be strengthened by applying all baselines across all datasets. Additionally, why is the CoRelNet not included in the baselines? It seems like one of the most relevant existing approaches.

---

Minor points:

5) Equation (2): What does adding $pos$ to the position embeddings $m_k$ add for the relational reasoning module? It is not dependent on the input, and cannot be adjusted by the relational reasoning module as it is part of the pre-trained, frozen SlotAttention model. Thus, I would expect that it doesn't provide much useful information.

6) Equation (4): $m_k$ will implicitly contain some information on the object shape, since $attn$ is used in Eq (2). Do you think this could allow the model to circumvent the relational bottleneck, if $m$ would be processed by some more powerful non-linearities?

7) Line 115: "endowing OCRA with an explicity variable-binding mechanism". What do you mean with this statement?

8) line 67: what is a "position-wise fully-connected layer"? Are you referring to the 1x1 convolutions in Table S2? In general, I think it would be helpful to draw direct connections between the components of the architecture described in Table S2 and the components described in the main text.

9) Eq. (1) + (2): what is the $\cdot$ symbol referring to here? In Eq. (3), this symbol is used to describe a dot procuct, which would not make much sense here, I think.

10) Lines 255-262: I would move most of this description into the method section.

11) Table 2: Use the same order for the different tested conditions in the table as is used within the text.

---

[1] Jiang, J., Deng, F., Singh, G., & Ahn, S. (2023). Object-centric slot diffusion. arXiv preprint arXiv:2303.10834.

[2] Wu, Z., Dvornik, N., Greff, K., Kipf, T., & Garg, A. (2022). Slotformer: Unsupervised visual dynamics simulation with object-centric models. ICLR 2023

[3] Dittadi, A., Papa, S., De Vita, M., Schölkopf, B., Winther, O., & Locatello, F. (2021). Generalization and robustness implications in object-centric learning. ICML 2022

**Questions:**

As described above, I believe the main weakness of this paper is its current positioning relative to the existing research landscape. If the authors could improve this point, or strengthen the current claims by providing additional experiments, I would be happy to adjust my score.


**Limitations:**

While the relational bottleneck improves the model's ability to generalize to previously unseen objects, I would expect it to also limit the model's applicability to other settings. For example, in the experiments, a separate model is applied for each separate task and thus for each separate relation to be learned. I would expect that the relational bottleneck would harm the model's performance if it had to differentiate several relations at once. Additionally, I would expect it to be more difficult to represent more complex relations between objects, as the slightly worse results on the spatial relations in the SVRT dataset might indicate.

---

> ### Author Rebuttal · Authors · 2023-08-10
>
> We thank the reviewer for their helpful suggestions and comments. We provide a point-by-point reply below:
>
> 1. Positioning relative to previous work:
>     - We have endeavored to cite all relevant previous work in the introduction and related work sections. While it was not feasible to implement baselines based on each and every one of these references, we have also done our best to select a representative subset of baselines that capture the important computational features of this previous work, and that are likely to represent the strongest possible baselines in the context of the abstract visual reasoning tasks we consider. Specifically, the slot transformer baseline is extremely similar to models from the references cited on lines 46 and 135, and was modeled closely on the implementation of [1] (ref 25 in the paper). The interaction network in our slot-IN baseline was highly similar to the models from the references cited on line 170, and was based closely on [2] (ref 43 in the paper). We believe that these baselines capture the important properties of the referenced papers, but we would be happy to implement specific additional baselines at the reviewer’s suggestion.
> 2. Clarifying novel components of proposed model:
>     - The factorization of slot attention into separate spatial and non-spatial components (equations 1 and 2) is indeed a novel feature of our model. The relational embeddings described in section 2.2 also constitute a novel aspect of the model. Section 2.3 describes passing these relational embeddings to a transformer, which is a novel use of transformers. Overall, the architecture does employ some components from previous work (slot attention, transformers), but they are configured in a novel way to form a broader architecture with very different properties than the individual components alone.
> 3. Missing references:
>     - We thank the reviewer for bringing these references to our attention. We agree that they are highly relevant to this work, and have added them to the Related Work section, along with a few other missing references that we identified from [3].
> 4. Systematic evaluation of all baselines, and addition of CoRelNet baseline:
>     - We agree with the reviewer’s suggestion to more systematically evaluate all baselines on all tasks. We have now included this evaluation in the revised paper (see Figures 1-3 of attached PDF). **We found that OCRA performed better than the baseline models in most settings, with the best average performance across all tasks, and especially strong performance in the most extreme generalization regimes.**
>     - **We have also added a slot-CoRelNet baseline**, by combining slot attention with CoRelNet. This baseline did not perform very well, likely due to the random permutation of objects in slot attention.
>
> Minor points:
>
> 5. Role of positional embeddings in relational reasoning:
>     - The positional embeddings are used to index the objects over which relations are computed. In other words, our relational embeddings can be interpreted as representing information such as ‘the object at location $i$ has the relation $R$ with the object at location $j$’, where $i$ and $j$ are spatial locations, and $R$ is a relation such as ‘same shape’. These positional indices are necessary to keep track of which relations correspond to which objects.
> 6. Potential to break bottleneck through position embeddings:
>     - It is very important to our formulation that $m_k$ be a *linear* combination of the position embeddings. A nonlinear transformation of the position embeddings could indeed result in the extraction of implicit shape information, and therefore break the relational bottleneck. We have added this important clarification to our description of the model.
> 7. Meaning of ‘explicit variable-binding mechanism’:
>     - This refers to the fact that our relational embeddings use positional information as indices, as described in the response to point #5 above (i.e., the relations are bound to explicit indices for the corresponding objects).
> 8. 'Position-wise fully-connected layer' vs. '1x1 convolutions':
>     - Yes these refer to the same components. We agree this is confusing, and have revised the paper to consistently use the term ‘1x1 convolutions’.
> 9. Clarification about equations 1 and 2:
>     - Thanks to the reviewer for catching this mistake. These are simply matrix multiplications. They have been reformatted as follows (where $\boldsymbol{attn}_{T} \in \mathbb{R}^{K\times N}$, $\operatorname{flatten}(\boldsymbol{feat}) \in \mathbb{R}^{N\times D}$, and $\operatorname{flatten}(\boldsymbol{pos}) \in \mathbb{R}^{N\times D}$):
>
>         $z_{k} = \boldsymbol{attn}_{T} \operatorname{flatten}(\boldsymbol{feat})[k]$
>
>         $m_{k} = \boldsymbol{attn}_{T} \operatorname{flatten}(\boldsymbol{pos})[k]$
>
> 10. Moving lines 255-262:
>     - We have now moved these lines to the supplementary.
> 11. Ordering of entries in Table 2:
>     - We have reordered the table accordingly.
>
> Limitations:
> - It is an interesting question the extent to which the proposed model could simultaneously model multiple relations. In future work, we plan to investigate how the model performs in multi-task settings. One possibility is to develop a 'multi-head' version of the current model, which may be better suited to this setting. We have added some discussion of this issue to the revised paper.
> - In the case of the SVRT spatial relation tasks, the model's performance was primarily limited by suboptimal object segmentation in our slot attention module. This was partly remedied by training slot attention for longer (see updated results in attached PDF). But we agree that it would be interesting in future work to evaluate the model on more complex real-world relations.
>
> [1] Mondal et al. (2023). Learning to reason over visual objects.
>
> [2] Watters et al. (2017). Visual interaction networks: Learning a physics simulator from video.
>
> [3] Jiang et al. (2023). Object-centric slot diffusion.

---

> > ### Comment · Reviewer_6gd3 · 2023-08-14
> >
> > Thank you for your response.
> >
> > I propose that the authors incorporate their responses to points 1 and 2 into their paper. I believe that doing so will better highlight its contributions. Moreover, the limitations I previously noted could enrich the paper if acknowledged explicitly. All my other concerns have been addressed adequately, and I will update my rating to "weak accept".

---

> > > ### Author Response · Authors · 2023-08-14
> > > **Further revisions**
> > >
> > > We thank the reviewer for these suggestions. We agree that adding additional discussion of these issues would further improve the paper. To address this, we have made the following revisions:
> > >
> > > - At the end of Related Work, we have added the following statement clarifying the connection between the baselines and previous work (Note that the references are those that were cited on lines 135 and 170 of the original submission):
> > >
> > >     ‘To empirically evaluate the importance of the relational bottleneck, we compare our approach with a set of baseline models that capture the key computational properties of previous models, including a slot-transformer baseline that combines slot attention with a transformer reasoning module [9, 34, 46, 25], and a slot-interaction-network baseline that combines slot attention with an interaction-network reasoning module [43, 39, 15, 47, 2, 19, 35, 41, 12, 36].’
> > >
> > > - We have also added the following statement to the end of Related Work, to clarify the novel aspects of our proposed approach:
> > >
> > >     ‘To summarize our contributions relative to previous work, our proposed model includes:
> > >
> > >      1. A novel object representation format that is factorized into distinct feature and position embeddings (Equations 1 and 2), enabling a form of explicit variable-binding.
> > >
> > >     2. A novel relational embedding method that implements a relational bottleneck (Equations 3 and 4).
> > >
> > >     3. An architecture that combines these elements with preexisting components (slot attention and transformers) in a novel manner to support systematic visual reasoning from complex multi-object visual displays.’
> > >
> > > - Finally, we have now revised the Conclusion to more explicitly address the limitations identified in this and other reviews:
> > >
> > >     ‘6. Limitations and Future Directions
> > >
> > >     In the present work, we have presented a model that integrates object-centric visual processing mechanisms (providing the ability to operate over complex multi-object visual inputs) with a relational bottleneck (providing a strong inductive bias for learning relational abstractions that enable human-like systematic generalization of learned abstract rules). Though this is a promising step forward, and a clear advance relative to previous models of abstract visual reasoning, it is likely that scaling the present approach to real-world settings will pose additional challenges. First, real-world images are especially challenging for object-centric methods due to a relative lack of consistent segmentation cues. However, there has recently been significant progress in this direction [33], in particular by utilizing representations from large-scale self-supervised methods [50, 7], and it should be possible to integrate these advances with our proposed framework. Second, the current approach assumes a fixed number of slot representations, which may not be ideal for modeling real-world scenes with a highly variable number of objects [51]. Though we did not find that this was an issue in the present work, in future work it would be desirable to develop a method for dynamically modifying the number of slots. Third, OCRA’s relational operator is applied to all possible pairwise object comparisons, an approach that may not be scalable to scenes that contain many objects. In future work, this may be addressed by replacing the transformer component of our model with architectures that are better designed for high-dimensional inputs [16]. Finally, it is unclear how our proposed approach may fare in settings that involve more complex real-world relations, and settings that require the discrimination of multiple relations at once. It may be beneficial in future work to investigate a ‘multi-head’ version of our proposed model (analogous to multi-head attention), in which multiple distinct relations are processed in parallel. A major challenge for future work is to develop models that can match the human capacity for structured visual reasoning in the context of such complex, real-world visual inputs.’

---

### Official Review · Reviewer_HzsY · 2023-07-04

**Soundness:** 4 excellent
**Presentation:** 4 excellent
**Contribution:** 4 excellent
**Rating:** 7
**Confidence:** 3

**Summary:**

- The paper seeks to tackle various visual reasoning problems with a focus on systematic generalization.
- The paper argues that we need explicit inductive bias to extract object-object relationships and express these relationships via a low-capacity representation.
- The paper seeks to do this by first extracting object vectors (or slots) using pre-trained slot attention. It then performs a dot-product of these slots to obtain a “relation embedding” — which is really a scalar number expressed as an embedding (as far as I understand). These relation embeddings are then given to a transformer and the whole model (except the slot attention) is trained to classify whether the given input is conformant to the true pattern or not.
- The performance is then evaluated on a systematically OOD test set and compared with various baselines and model ablations on 3 visual reasoning benchmarks: ART, SVRT, and CLEVR-ART. Here, CLEVR-ART is a novel dataset proposed within this paper.

**Strengths:**

1. Simple and elegant architecture.
2. Propose a new dataset for visual reasoning and systematic generalization called CLEVR-ART, a sort of visually complex variant of ART. This appears to be a useful contribution to facilitating progress in the community.
3. Good systematic generalization performance compared to baselines.
4. Several useful baseline comparisons. For instance, the paper shows that a standard transformer when applied to slots is not enough to systematically generalize and the proposed relation layer is useful.
5. Useful ablations. For instance, the paper shows that all of the following are useful: relation bottleneck, relation embedding, object-centric inputs, decomposing position and appearance, etc.

**Weaknesses:**

I have several questions and possible avenues for improvements that I describe in the “Questions” section.

**Questions:**

1. Also, what part of the architecture is enforcing the “relational bottleneck”? Is it due to the fact that the dot product of two slots results in a scalar number? If yes, then this should be highlighted somehow both in the text and perhaps also in Figure 1. One may also conduct an experiment in which the “bottleneck” dimension is gradually increased (e.g., 1 to 2 to 5 and so on) and show that this gradually worsens the generalization performance.
2. Line 30: Regarding learning relational abstractions: Is it possible to take a large batch of inputs, extract relation “embedding” and visualize object pairs that have similar relational “embedding”? This would give more credence to the fact that relational abstractions are indeed being inferred.
3. Line 108: Is it necessary to say “shared” here? IMO, I wouldn’t have assumed that the projection matrices were not shared (based on the equations).
4. It should be better highlighted in the introduction section and the abstract that CLEVR-ART is a novel contribution. This should also be discussed in the related work relative to the existing visual reasoning benchmarks. It would also be useful to make a statement about whether this dataset will be released to the research community or not.
5. Table 1: Why are several baselines shown for ART not shown for CLEVR-ART and SVRT? I think these baselines will be useful to show for all datasets. Also, I would suggest using a consistent format for reporting the results of all 3 datasets i.e., picking either the bar-plot format or the tabular format.
6. L263: In the case of inputs involving multiple images, how are they processed by the network? As I understand, the model does not sequentially consume multiple images. Does “inserted” mean programmatically generating the candidate images, each containing multiple objects?
7. L283-298. I find the discussion of the results rather small. For instance, what is the rationale for the comparison with ESBN or GAMR? What is the key distinguishing characteristic of those baselines with respect to the proposed one? What can we learn from the result of this comparison i.e., why does the proposed model outperform?
8. (Minor) Line 34: Would be good to cite the paper(s) that support this statement “By biasing architectures to process visual inputs in terms of relations between objects, these recent approaches have achieved strong systematic (i.e., out-of-distribution) generalization of learned abstract rules, given only a small number of training examples.”

I am giving a score of 6 in the hope that the authors will try and address some of my questions and concerns. If addressed to some degree, I will happily increase the score.

**Limitations:**

Yes, the limitations are discussed in the last line of the conclusion.

---

> ### Author Rebuttal · Authors · 2023-08-10
>
> We thank the reviewer for their helpful suggestions and comments. We provide a point-by-point reply below:
>
> 1. Mechanism enforcing the relational bottleneck:
>     - The reviewer asks whether the dot product enforces a relational bottleneck only because it compresses inputs to a single dimension. To investigate this, we performed an ablation experiment, in which the dot product was replaced with a linear layer that projected the concatenated pair of input embeddings to a single dimension. We found that this model performed very poorly on the $m=95$ (most difficult) generalization regime for the RMTS (50.7% test accuracy) and ID (48% test accuracy) tasks. Thus, the relational bottleneck does not result from the unidimensional nature of the dot product per se. Instead, we believe that the most important feature of the dot product is that it is inherently relational, in that it is based on a multiplicative interaction between the two input embeddings. By contrast, when using a linear layer (or MLP, as in the ‘- Relational Bottleneck’ ablation model) to model relations, there is nothing to prevent the model from simply copying individual object features into the ‘relational’ embedding (thus making it not really relational). We have included the result of this ablation experiment in the revised paper, along with a discussion of the implications (we are unfortunately not able to include the specific revisions in this reply due to space constraints).
> 2. Visualization of relational embeddings:
>     - This is an excellent suggestion. We regrettably did not have time to carry out this analysis during the rebuttal period, but we will try to do so before the end of the discussion period and share the results here time permitting.
> 3. Specification of ‘shared’ weight matrices:
>     - We agree that the ‘shared’ nature of the weight matrices (mentioned in line 108) most likely doesn’t need clarification. However, our concern was that some readers may expect these to be separate matrices, in line with the common use of separate ‘key’ and ‘query’ matrices in self-attention. We included this clarification to avoid that potential confusion.
> 4. Clarification about novelty of CLEVR-ART and release to public:
>     - We have added the clarifications to the abstract and introduction concerning the novelty of CLEVR-ART. We have also added discussion of CLEVR-ART relative to other visual reasoning benchmarks to the Related Work section (specific revisions omitted here due to space constraints).
>     - We do plan to make the dataset publicly available upon acceptance of the paper.
> 5. Systematic testing of baseline and consistent format for results:
>     - We thank the reviewer for these suggestions. We have now evaluated all baselines on all three of these benchmarks (with the exception of Attn-ResNet, for which we do not have source code). The results can be found in Figures 1-3 of the attached PDF. **We found that OCRA performed better than the baseline models in most settings, with the best average performance across all tasks, and especially strong performance in the most extreme generalization regimes.** These results will be included in the final version of the paper, and we will also make sure to report all of these results as figures.
> 6. Evaluation on multi-object inputs:
>     - The reviewer is correct that our approach involved ‘programmatically generating the candidate images, each containing multiple objects’. We have described this approach in line 199 of the original submission: ‘*As originally proposed, the ART dataset involved pre-segmented visual objects. Here, we investigated a version of this task involving multi-object visual displays (see Supplementary Section S1 for details).*’
> 7. Limited discussion of results:
>     - We agree that more discussion of the results would be informative. Briefly, many of the baseline models do not contain a relational bottleneck (Transformer, IN, RN, GAMR, ResNet), and their ability to generalize is therefore limited to iid settings (e.g., low values of $m$ for the ART tasks). The comparison with these models is thus intended to evaluate the importance of the relational bottleneck, which enables stronger performance in the OOD setting. The ESBN and CoRelNet architectures *do* include a relational bottleneck, but they are not designed with multi-object inputs in mind, and they therefore perform poorly on these tasks in all regimes (even when combined with our slot attention module). This is due primarily to the random permutation of slots in slot attention, which motivated our positional embedding variable-binding scheme (to keep track of which relations correspond to which pairs of objects). We have added discussion of these issues to the Results section (specific revisions omitted here due to space constraints).
> 8. Citing papers to support statement about OOD generalization:
>     - This statement is referring to the same papers cited at the beginning of the paragraph (line 30). We have added references to the revised paper to clarify this.

---

> > ### Comment · Reviewer_HzsY · 2023-08-18
> > **Thank You**
> >
> > Thank you for the response! I appreciate the response that it is the dot product that is playing the key role as evidenced by the failure of the ablation in which slot pairs were concatenated and mapped to single-dimension. Although I find this outcome very intriguing and interesting, I am not fully clear why it does work. Is there something inherent about multiplicative interaction that leads to this? For instance, would the outer product also work similarly?
> >
> > Another question is: Consider that $\mathbf{z}$ contains the appearance information but not spatial position information while $\mathbf{m}$ contains spatial position information but not appearance information. Then, I think that Eq. 4 should lose information about which object appearance is associated with which spatial position. Is this true and is this by design?
> >
> > I also read other reviewers' comments, especially the one asking for the CoRelNet baseline. I think this was a valid point. I agree that this baseline should have been compared. I appreciate the authors' addition of CoRelNet results and I also agree with the authors' response that CoRelNet should struggle to handle permutation invariant representations e.g., slots of slot attention.
> > However, since I missed noticing this related work the first time, I am now less confident that I fully grasp all pieces of the related work. As such, I will reduce my confidence from 4 to 3. But I still maintain my rating of 6 to the best of my understanding.

---

> > > ### Author Response · Authors · 2023-08-18
> > >
> > > Thank you for the continued engagement. We have now carried out the analysis that was suggested in question 2 of the initial review (visualization of relational embeddings). We performed this analysis specifically for the same/different task, in the $m=95$ generalization regime. This was the most difficult generalization regime in our task, so we thought this would be the strongest test of whether the relational embeddings truly captured abstract relations. We performed PCA on the relational embeddings (the output of equation 3 in the paper) for a set of 100 problems, and visualized the results along the first two principal components. The results formed two distinct clusters, one for pairs of objects with the same shape, and one for pairs of objects with different shapes. Thus, the relational embeddings identified the relevant relational abstraction (same/different), despite the fact that the inputs involved completely novel shapes not observed during training. We will include these results in the final version of the paper, and thank the reviewer for this suggestion.
> > >
> > > Inner vs. outer product: This is a great question. To test it, we performed an additional ablation experiment, in which the dot product in the relational operator (equation 3 of the paper) was replaced with an outer product. Note that this results in a matrix of size $D \times D$. We flattened this matrix, and passed it through a learned linear layer to reduce it to size $D$. This version of the model did not perform nearly as well as our primary model (see results in Table below). This suggests that our relational operator depends on *both* multiplicative interaction *and* compression. Ablation models that only have compression (the ablation model with a learned linear projection to one dimension) or only have multiplicative interaction (the new ablation model involving an outer product) do not perform as well. We will add these results and some additional discussion of these issues to the final version of the paper. We thank the reviewer for raising these issues, as we feel that they help to clarify the important factors underlying the implementation of the relational bottleneck.
> > >
> > > | | RMTS | ID |
> > > | -- | -- | -- |
> > > | OCRA (original) | **85.31 ± 2.0** | **92.80 ± 0.3** |
> > > | OCRA (outer product) | 62.84 ± 1.6 | 69.58 ± 1.1 |
> > >
> > > Clarification about equation 4: In equation 4, information about the appearance of *individual* objects is discarded, but information about the *relationship* between the appearance of two objects is preserved, and this is then associated with the spatial position of the two objects. The discarding of information about individual object appearance is indeed by design. Please let us know if we can provide further clarification.
> > >
> > > Regarding the related work, we would be happy to discuss any remaining issues that we can help clarify.
> > >
> > > We have made our best effort to address all of the concerns raised in the initial reviews and during the discussion period. If there are any issues that haven’t been sufficiently addressed, we would be happy to discuss further.

---

> > > > ### Comment · Reviewer_HzsY · 2023-08-18
> > > > **Thank You!**
> > > >
> > > > The results are indeed very interesting. I look forward to seeing the said visualization. I raise my score to 7.

---

### Official Review · Reviewer_1vcp · 2023-07-07

**Soundness:** 3 good
**Presentation:** 3 good
**Contribution:** 2 fair
**Rating:** 6
**Confidence:** 3

**Summary:**

The research topic of this study is the development of a learning machine that achieves systematic generalization in reasoning over complex relations of objects in a visual input (still image). Toward this goal, the authors proposed a new neural network model (OCRA) taking inspirations from the recent results on effective inductive biases for systematic generalization in relational reasoning and methods for obtaining object-centric representations. More concretely, the proposed model comprised of three core components: first component (slot attention mechanism) to extract object-centric representation from a visual input containing multiple objects, the second component to compute pairwise relation embeddings, and the third component (transformer) to provide the final output related to the higher-order relations. The effectiveness of the proposed model was tested with three visual reasoning tasks, two existing and one new, and also compared with various baseline models. As a whole, the proposed model can be said the best in terms of systematic generalization among the compared models.  An ablation study was also conducted to evaluate the roles of the components in the proposed model and pretraining.

**Strengths:**

The top-level idea behind the proposed model, that is, combining the inductive biases for relational reasoning and a method for obtaining object-centric representations is clear and reasonable. How to implement the idea with the three core components are explained fairly well. Although the top-level idea might look somewhat straightforward at first glance given the advancement in the two directions, the concrete implementation is not trivial, and the differences from the existing work are described in the paper.

The effectiveness of the proposed model was tested with fairly rich experiments. Two existing task (ART and SVRT, both created with 2D shapes) and a new task developed based on the CLEVR (CLEVR-ART,  created with 3D shapes) were used and various baseline models including a very recent one (GAMR, accepted at this year's ICLR) are compared with the proposed model. An ablation study gives additional value to this work.

Development of a learning machine that achieves systematic generalization in complex visual reasoning is an important topic in AI. Although there is still a distance to the real-world applications as stated in Section 6 (Conclusion and Future Directions), the proposed method and the evaluation results will be of interest to the NeurIPS audience.

**Weaknesses:**

1. A weak point of this submission is lack of the source code of the proposed model as Supplementary Material. It is also unkown whether the source code will be made publicly available if the paper get accepted. These points cast a shadow on the reproducibility.

1. A relatively weak point of the proposed model itself seems to be around the number of slots, $K$. First, as stated in line 117, the proposed model computes all $K^2$ pairwise relations. Second, in the current model, $K$ should be defined in advance. These characteristics can be obstacles in the reasoning over the relations of objects in the real-world visual inputs. Namely, the fist point might affect the applicability (scalability) to complex natural images and there should be some additional mechanism to determine an appropriate $K$ in the first place if the current structure of the proposed model is kept.

1. (Related to 2.) Although it is stated that there exists gap between the problems addressed in this study and the real-world vision in Section 6 (Conclusion and Future Directions), the details are not explained.




**Questions:**

Questions

1. Can the source code of the proposed method be provided at the Author Rebuttal period? In addition, will the source code made publicly available if the paper get accepted?

1.  What are your thoughts on the second point in Weaknesses section above?

Suggestion

It would be beneficial if the authors could add detailed descriptions about what kind of differences between the problems treated in this study and the real-world vision should be overcome, and also add explanations about the relation between those differences and  self-supervised learning methods mentioned in Section 6 if possible.

**Limitations:**

The authors stated in Section 6 (Conclusion and Future Directions) that the problems addressed in this study are still simple compared with the the real-world vision.  However, currently it is not explained in details what kind of differences between the problems treated in this study and the real-world vision should be overcome. Please also refer to the Weaknesses and Questions sections above.

---

> ### Author Rebuttal · Authors · 2023-08-09
>
> We thank the reviewer for their helpful suggestions and comments. We provide a point-by-point reply below:
>
> 1. Lack of source code:
>     - We thank the reviewer for bringing this to our attention. **We have now provided the AC with an anonymized link to the source code for our model. We will also make this code publicly available upon acceptance of the paper.**
> 2. Fixed number of slots and scalability:
>     - A desirable feature of slot attention is that the number of slots can be set to the maximum number of expected objects, and slots that don’t correspond to objects will remain largely unutilized (e.g. if there are 6 slots but only 2 objects in an image, typically only 2 slots will end up being used). OCRA inherits this property from slot attention. Therefore, in the ART dataset, in which different problem types contain different numbers of objects (ranging between 2 and 6), we set the number of slots to $K=6$ for all problems. However, we agree that an ideal approach would involve autonomously adjusting the slots based on the number of objects present in an image. We have added some additional discussion of this issue to the paper (see excerpt from revised text below).
>     - We also agree that computing all pairwise relations may pose a challenge when scaling to real-world scenes with many objects. In future work, we believe that this may be addressed by replacing the transformer component of our model with other methods that are better equipped to deal with very high-dimensional inputs, e.g. [1]. We have added additional discussion of this issue to the revised paper (see excerpt below).
> 3. Lack of detail on obstacles to scaling:
>     - We agree that this issue deserves further attention.  We have revised the discussion of these issues in the conclusion of the paper as follows (new text in bold, note also reference numbers are not the same as those in the paper):
>
>         ‘In the present work, we have presented a model that integrates object-centric visual processing mechanisms (providing the ability to operate over complex multi-object visual inputs) with a relational bottleneck (providing a strong inductive bias for learning relational abstractions that enable human-like systematic generalization of learned abstract rules). Though this is a promising step forward, and a clear advance relative to previous models of abstract visual reasoning, it is likely that scaling the present approach to real-world settings will pose additional challenges. **First, real-world images are especially challenging for object-centric methods due to a relative lack of consistent segmentation cues. However, there has recently been significant progress in this direction [2], in particular by utilizing representations from large-scale self-supervised methods [3], and it should be possible to integrate these advances with our proposed framework. Second, the current approach assumes a fixed number of slot representations, which may not be ideal for modeling real-world scenes with a highly variable number of objects [4]. Though we did not find that this was an issue in the present work, in future work it would be desirable to develop a method for dynamically modifying the number of slots. Finally, OCRA’s relational operator is applied to all possible pairwise object comparisons, an approach that may not be scalable to scenes that contain many objects. In future work, this may be addressed by replacing the transformer component of our model with architectures that are better designed for high-dimensional inputs [1].**’
>
> [1] Jaegle et al. (2021). Perceiver: General perception with iterative attention.
>
> [2] Seitzer et al. (2022). Bridging the gap to real-world object-centric learning.
>
> [3] Caron et al. (2021). Emerging properties in self-supervised vision transformers.
>
> [4] Zimmermann et al. (2023). Sensitivity of Slot-Based Object-Centric Models to their Number of Slots.

---

> > ### Comment · Reviewer_1vcp · 2023-08-18
> >
> > Thank you very much for thoroughly considering my review. All of my questions and suggestions have been adequately responded. I raised the score for Presentation from 2 to 3 and that for Rating from 5 to 6.
> >
> > \# The link to the source code was shared by AC.

---

### Official Review · Reviewer_bzvn · 2023-07-13

**Soundness:** 2 fair
**Presentation:** 3 good
**Contribution:** 2 fair
**Rating:** 6
**Confidence:** 3

**Summary:**

This work proposed a new method, named OCRA, that combines object-centric presentation learning (for object abstraction) and a relational bottleneck (for relational abstraction). Particularly, OCRA consists of three components: 1) a slot attention to extract object level representations, 2) a relational operator to get pairwise visual relations, and 3) a transformer to model higher-order relations. The slot attention model is pretrained on a large dataset and the relational modules are trained on a small task-specific dataset while freezing the slot attention. Experiments were performed on three datasets: ART, SVRT and CLEVR-ART to show the effectiveness of the proposed method.

**Strengths:**

1. The idea of combining slot attention with a relational bottleneck (relation operator and transformer) for solving visual relational reasoning problems sounds interesting and also novel to me.
2. The presentation of the idea and the overall writing are very clear.
3. Experiments on synthetic datasets (ART and SVRT and CLEVR-ART) shows the proposed method can work for various relational reasoning tasks and it achieves better performance than baselines in many cases.

**Weaknesses:**

1. For the benchmarks, I have a concern about their simplicity. For example, the performance of many methods on the ART dataset is close to 100%. Similar observations also exist in other two datasets. Does it mean we already have achieved human-level visual reasoning performance or we need better benchmarks to evaluate the success of methods? I think considering some more challenging benchmarks will make the results more convincing. For example, RAVEN [1] and Bongard-HOI [2] are two challenging benchmarks for testing a model's visual relational reasoning abilities. In particular, Bongard-HOI considers the real-world natural images.
2. In Figure 4, it is a little surprising that OCRA performs worse than ResNet on SVRT - spatial relations with 1000 training examples. Any intuition on this?
3. For the ablation studies, to test the impact of slot attention, it is too naive to do “feature map divided into a 4x4 grid”. How about comparing slot attention with some standard SSL trained object-centric representation learning methods? Also, from Table 2, we can see that Transformer has a much higher impact than the slot attention and relational bottleneck on both RMTS and ID. Does it mean the most important part of OCRA is the transformer rather than slot attention and relational bottleneck? If so, it downplays the method's significance a little, in my opinion.

[1] RAVEN: A Dataset for Relational and Analogical Visual rEasoNing, CVPR 2019.

[2] Bongard-HOI: Benchmarking Few-Shot Visual Reasoning for Human-Object Interactions, CVPR 2022.

**Questions:**

My major concerns and questions are in how the experiments support the effectiveness and significance of the proposed method. Please see the weaknesses part for more details.

**Limitations:**

The authors have well addressed the limitations.

---

> ### Author Rebuttal · Authors · 2023-08-09
>
> We thank the reviewer for their helpful suggestions and comments. We provide a point-by-point reply below:
>
> 1. Synthetic vs. real-world benchmarks:
>     - The primary focus of the benchmarks we consider is on the more extreme **out-of-distribution generalization and low-sample regimes**. In the most extreme generalization regime of the ART dataset ($m=95$ in Figure 1 of the attached PDF above), **most baselines perform very poorly, with many displaying near-chance performance**. The baseline performance on CLEVR-ART and the same/different SVRT tasks (especially in the low-sample regime, with only 500 training examples) is similarly poor. Thus, though these tasks don’t involve real-world images, they are nevertheless extremely challenging tests of out-of-distribution visual reasoning.
>     - We also agree with the reviewer that it is important to extend models of visual reasoning so that they are capable of processing real-world visual inputs. We believe that this work takes a significant step in that direction, by proposing a model capable of strong **OOD generalization from multi-object inputs (ART and SVRT) and even photorealistic three-dimensional objects (CLEVR-ART)**. We agree that it will be important in future work to take this even further, by evaluating the model on real-world images, utilizing datasets such as Bongard-HOI. Though this poses a few challenges (which we discuss in the general response above), we do not see any fundamental obstacles to extending our model to these kinds of datasets, and plan to do so in future work.
> 2. Relatively poor performance on SVRT - spatial relations:
>     - We found that OCRA’s poor relative performance on these tasks was related to the fact that the learned slot representations were not perfectly object-centric. This was partially remedied by training slot attention for longer. With further pretraining of slot attention, OCRA outperforms ResNet in the low-sample regime (500 training examples), and performs on par with ResNet in the high-sample regime (1k training examples; see Figure 3 in the attached PDF above). Note also that OCRA’s performance on these tasks is generally strong (around 90% test accuracy), and higher than it is for the same/different tasks (which were extremely challenging for many of the baselines).
> 3. Testing impact of slot attention and relational bottleneck:
>     - To better test for the impact of slot attention, the reviewer proposes that we compare OCRA with a ‘standard SSL trained object-centric representation learning method.’ We are not sure which specific methods the reviewer has in mind. The purpose of the ‘- Slot Attention’ ablation is precisely to test for the impact of using object-centric representations, so this ablation model should not be object-centric. However, we agree that it is informative to compare to other self-supervised representation learning methods. We have now added a comparison with a state-of-the-art masked autoencoder model (MAE; using the code base from [1]) on the identity rules and distribution-of-3 ART tasks (it is not clear how to formulate the same/different and RMTS tasks in a generative manner). We applied this model to our tasks by masking out the final object in each problem (in the bottom right cell of the input), and training the model to fill in this patch. To select from the set of multiple choices, we then compared the model’s generated output with the four answer choices, and selected the choice with lowest mean-squared error. We found that this model performed considerably worse than OCRA, especially in the most extreme generalization regimes ($m=95$). We have included the results in a table below, and will also add them to the revised paper. If the reviewer had other methods in mind, we would be happy to test these and compare with our model if time permits.
>     - The reviewer notes that the most impactful ablation is the ‘- Transformer’ model. This demonstrates that higher-order relations (relations between pairwise relations, which are extracted by the transformer in OCRA) are essential to the model’s performance. However, we note that ablating either slot attention or the relational bottleneck also has a severe effect on performance. For instance, in the RMTS task, performance drops by ~29% without slot attention, and by ~22% without a relational bottleneck. This demonstrates that all major elements of the model (object-centric representations, relational abstraction, higher-order relations) play an important role.
>
>
> | Distribution-of-3 | $m=0$ | $m=50$ | $m=85$ | $m=95$ |
> | ------- | ------ | ------- | ------- | ------- |
> | OCRA (ours) | **98.86 ± 0.2** | **97.87 ± 0.2** | **96.09 ± 0.4** | **86.42 ± 1.2** |
> | MAE | **99.99 ± 0.0** | 56.47 ± 1.1 | 40.90 ± 1.2 | 28.85 ± 0.9 |
>
> | Identity rules | $m=0$ | $m=50$ | $m=85$ | $m=95$ |
> | ------- | ------ | ------- | ------- | ------- |
> | OCRA (ours) | **99.01 ± 0.0** | **98.01 ± 0.1** | **96.67 ± 0.2** | **92.80 ± 0.3** |
> | MAE | 66.55 ± 0.2 | 44.96 ± 0.6 | 38.39 ± 0.9 | 31.56 ± 1.0 |
>
> [1] He et al. (2022). Masked autoencoders are scalable vision learners.

---

> > ### Comment · Reviewer_bzvn · 2023-08-19
> > **Thanks**
> >
> > Thanks to the authors for providing detailed responses to my questions and adding the new experiments, which support the effectiveness of the slot attention. The rebuttal has addressed most of my concerns. I have also read other reviews and increased my rating to weak accept.

---

### Author Rebuttal · Authors · 2023-08-09

We would like to thank the reviewers for their many insightful suggestions and comments. We have added a number of new experiments to address the concerns raised, and revised the paper to improve clarity and provide further elaboration on some issues. We believe the paper is significantly improved as a result of these changes. Here, we will address the major issues raised by the reviewers, but we also provide point-by-point replies to each reviewers’ comments below.

- More systematic evaluation of baselines: We thank the reviewers for raising this issue, and have now included results for all baselines on all tasks, including six new baseline results for CLEVR-ART and SVRT, and one new baseline result for ART (see figures 1-3 in the attached PDF). These results include the implementation of a new baseline model ‘slot-corelnet’, that combined our pre-trained slot attention module with the corelnet [1] architecture. **We found that OCRA performed better than the baseline models in most settings, with the best average performance across all tasks, and especially strong performance in the most extreme generalization regimes.** Please note that we were not able to evaluate the ‘Attention-ResNet’ baseline on other tasks besides SVRT, as the SVRT result was reproduced from an earlier paper [2], for which no source code was provided.
- Lack of source code in our original submission: **We have now provided the AC with an anonymized link to the source code for our model. We will make this code publicly available upon acceptance of the paper. We will also make the CLEVR-ART dataset publicly available upon acceptance.**
- Concern about simplicity of benchmarks: The primary focus of the benchmarks we consider is **out-of-distribution generalization**. In the case of ART, performance of the baselines in the most extreme generalization regime ($m=95$ in Figure 1 of the attached PDF) is far from ceiling, with most performing at or near chance, whereas OCRA achieves a score of 88%. Performance of the baselines on the CLEVR-ART task (which also involves a significant degree of OOD generalization) and SVRT same-different tasks is similarly poor whereas OCRA achieves a score of 85% on both. Thus, despite the relative visual simplicity of the tasks that we consider, they are nevertheless an extremely challenging test of generalization and abstraction for baselines, whereas OCRA performs well.
- Importance of extension to real-world visual inputs: We do however agree that it is important to extend models of visual reasoning so that they are capable of processing real-world visual inputs, and indeed that goal is precisely what motivated the present work. Toward that end, we focused in this work on extending abstract visual reasoning methods to deal with multi-object visual inputs. This was also the motivation for developing the CLEVR-ART dataset, which combines OOD generalization and photorealistically rendered three-dimensional objects. In future work, we agree that it would be extremely valuable to take this even further, by evaluating models on OOD generalization in the context of real-world images (e.g. on the Bongard-HOI benchmark [3], see next comment for a discussion of the prospects for doing this).
- Lack of detail on differences between current benchmarks and real-world vision: The primary challenge that we envision in extending this work to more complex settings is the difficulty involved in extracting object-centric representations from real-world images. This is a challenging problem because real-world images contain less consistent segmentation cues than in synthetic tasks like CLEVR. However, there has recently been significant progress in this direction [4], and it should be possible to combine these advances with our proposed model, which is an important goal for future work. Another potential challenge is that real-world scenes often involve a large number of objects, which would yield an exponentially larger number of pairwise relations in our model. We believe that this could be addressed by replacing the transformer component with methods that are better designed for higher-dimensional inputs, e.g. [5]. Thus, although there are certainly challenges to be addressed, we do not envision any fundamental obstacles to further scaling of the proposed approach. We have added these clarifications and additional discussion points to the revised manuscript.

[1] Kerg et al. (2022). On neural architecture inductive biases for relational tasks.

[2] Vaishnav & Serre (2023). GAMR: A guided attention model for (visual) reasoning.

[3] Jiang et al. (2022). Bongard-HOI: Benchmarking few-shot visual reasoning for human-object interactions.

[4] Seitzer et al. (2022). Bridging the gap to real-world object-centric learning.

[5] Jaegle et al. (2021). Perceiver: General perception with iterative attention.

---

### Decision · Program_Chairs · 2023-09-21

**Decision:**

Accept (poster)

**Comment:**

This paper proposes ORCA, a model for object-centric relational abstraction and reasoning for visual data. The method combines Slot Attention with a relational reasoning component and proves effective for several challenging synthetic visual reasoning tasks. The paper further introduces a new reasoning task based on CLEVR, called CLEVR-ART, which presents a valuable contribution to this area.

The reviewers agree that this is a paper of high quality that presents a valuable contribution to the field. While the considered benchmark tasks are visually simplistic, the experimental evaluation is very thorough with a solid range of baseline comparisons and model ablations. While the positioning to related work could be expanded, this is a solid submission that in my view meets the bar for acceptance at NeurIPS.